

**The pan-tropical response of soil moisture to El Niño**
Kurt C. Solander[1], Brent D. Newman[1], Alessandro Carioca de Araujo[2], Holly R. Barnard[3], Z. Carter
Berry[4], Damien Bonal[5], Mario Bretfeld[6,13], Benoit Burban[7], Luiz Antonio Candido[8], Rolando Célleri[9],
Jeffery Q. Chambers[10], Bradley O. Christoffersen[11], Matteo Detto[12,13], Wouter A. Dorigo[14], Brent E.
Ewers[15], Savio José Filgueiras Ferreira[8], Alexander Knohl[16], L. Ruby Leung[17], Nate G. McDowell[17],
Gretchen R. Miller[18], Maria Terezinha Ferreira Monteiro[19], Georgianne W. Moore[20], Robinson
Negron-Juarez[10], Scott R. Saleska[21], Christian Stiegler[16], Javier Tomasella[22], Chonggang Xu[1]
[1] Earth and Environmental Sciences, Los Alamos National Laboratory, Los Alamos, NM
[2] Brazilian Agricultural Research Corporation, Embrapa Amazônia Oriental, Manaus, Brazil
[3] Department of Geography, University of Colorado, Boulder, CO
[4] Schmid College of Science and Technology, Chapman University, Orange, CA
[5] Université de Lorraine, AgroParisTech, INRA, UMR Silva F-54000, Nancy, France
[6] Department of Ecology, Evolution, and Organismal Biology, Kennesaw State University, Kennesaw,
GA
[7] INRA, UMR EcoFoG, AgroParisTech, Cirad, CNRS, Université des Antilles, Université de Guyane,
Kourou, France
[8] Coordination of Environmental Dynamics, National Institute for Amazonia Research, Manuas,
Brazil
[9] Department of Water Resources and Environmental Sciences, University of Cuenca, Cuenca,
Ecuador
[10] Earth and Environmental Sciences, Lawrence Berkeley National Laboratory, Berkeley, CA
[11] Department of Biology, University of Texas Rio Grande Valley, Edinburg, TX
[12] Department of Ecology and Evolutionary Biology, Princeton University, Princeton, NJ
[13] Smithsonian Tropical Research Institute, Panama City, Panama
[14] Department of Geodesy and Geoinformation, Vienna University of Technology, Vienna, Austria
[15] Department of Botany, University of Wyoming, Laramie, WY
[16] Bioclimatology, University of Goettingen, Goettingen, Germany
[17] Atmospheric Sciences and Global Change, Pacific Northwest National Laboratory, Richland, WA
[18] Civil Engineering, Texas A&M University, College Station, TX
[19] Climate and Environment, National Institute for Amazonia Research, Manaus, Brazil
[20] Department of Ecosystem Science and Management, Texas A&M University, College Station, TX
[21] Ecology and Evolutionary Biology, University of Arizona, Tucson, AZ
[22] Coordination of Research and Development, National Centre for Monitoring and Early Warning of
Natural Disasters, Cachoeira Paulista, Brazil





**Abstract**
The 2015-16 El Niño event ranks as one of the most severe on record in terms of the
magnitude and extent of sea surface temperature (SST) anomalies generated in the tropical
Pacific Ocean.  Corresponding global impacts on the climate were expected to rival, or even
surpass, those of the 1997-98 severe El Niño event, which had SST anomalies that were
similar in size. However, the 2015-16 event failed to meet expectations for hydrologic change
in many areas, including those expected to receive well above normal precipitation. To better
understand how climate anomalies during an El Niño event impact soil moisture, we
investigate changes in soil moisture in the humid tropics (between $\pm25°$) during the three
most recent super El Niño events of 1982-83, 1997-98, and 2015-16, using data from the
Global Land Data Assimilation System (GLDAS). First, we validate the soil moisture estimates
from GLDAS through comparison with in-situ observations obtained from 16 sites across five
continents, showing an $r^2$ of 0.54. Next, we apply a k-means cluster analysis to the soil
moisture estimates during the El Niño mature phase, resulting in four groups of clustered
data. The strongest and most consistent decreases in soil moisture occur in the Amazon basin
and maritime southeast Asia, while the most consistent increases occur over east Africa. In
addition, we compare changes in soil moisture to both precipitation and evapotranspiration,
which showed a lack of agreement in the direction of change between these variables and
soil moisture most prominently in the southern Amazon basin, Sahel and mainland southeast
Asia. Our results can be used to improve estimates of spatiotemporal differences in El Niño
impacts on soil moisture in tropical hydrology and ecosystem models at multiple scales.






## Introduction


The El Niño Southern Oscillation (ENSO) is one of the major coupled ocean-
atmosphere modes of variability internal to the Earth system operating on interannual
timescales (Jones et al., 2001). ENSO refers to basin-wide changes in the air-sea interaction
associated with changes in the sea surface temperatures (SSTs) of the tropical Pacific region.
Depending on the directionally of the SST deviation, ENSO events are classified in two
modes—El Niño, or the warm mode, when unusually warm water exists in the eastern
tropical Pacific Ocean off the South American coast—and La Niña, or the cool mode, when
anomalously cool water pools exist in approximately the same location (Trenberth, 1997).
Associated impacts on weather and climate over terrestrial areas are variable but typically
strongest in the low-latitude and some of the mid-latitude regions of North and South
America, east Africa, Asia and Australia (Ropelewski and Halpert, 1989); however, the
influence of ENSO on weather and climate has been detected around the globe outside of
these regions through teleconnection (Iizumi et al., 2013). Although we bring up ENSO here
to highlight the mode duality of this climate feature, the focus of our study presented here is
solely on the El Niño mode of ENSO.
An important factor that controls the teleconnection in climate and weather patterns
caused by El Niño is the magnitude of the given El Niño event. Of the 39 El Niño events that
have occurred since 1952, those occurring in 1972-73, 1982-83, 1997-98 and 2015-16 are
categorized as "super El Niño" events (Hong et al., 2014). Although occurring at a much
lower frequency than a non-super El Niño event, these events account for a
disproportionately large amount of the global climate anomalies associated with El Niño.
There is debate as to whether or not the 2015-16 event can be classified as a super El Niño





based on the lack of specific features that characterize a super El Niño including strong far
east Pacific SST anomalies, unusually high global SSTs, reduced outgoing longwave radiation
(OLR), and weaker surface wind and sea surface height in the eastern Pacific (Hameed et al.,
2018). We use the definition put forth by Hong et al., (2014) that defines a super El Niño as
one with Niño-3 SST anomalies greater than one standard deviation above others in the
instrumental record (Trenberth, 1997), coupled with a Southern Hemispheric transverse
circulation that is robust relative to that of other El Niños. The 2015-16 event fits the super
El Niño classification using this definition (Huang et al., 2016; Chen et al., 2017).

Prediction of the climatic or hydrologic response over the land surface from an El

Niño has proved to be difficult even during a super El Niño event. For example, none of the
25 forecasts of precipitation patterns produced from various models could accurately
predict precipitation over the western US during the 2015-16 El Niño event (Wanders et al.,
2017). Indeed, Wanders et al., (2017) reported that less than half of the forecasts predicted
the directionality of precipitation changes correctly. An evaluation of the three most recent
super El Niños revealed that although drought during January to March (JFM) was
widespread over the entire Amazon basin during the 1982-83 and 1997-98 events, during
the 2015-16 event the western half of the basin actually got wetter (Jiménez-Muñoz et al.,
2016). The authors indicate that spatial differences in the SST anomaly during JFM 2015-16
relative to other super El Niños may have contributed to this anomaly (e.g. Yu and Zou,

2013).

Given the diversity of El Niño impacts on precipitation, it is not clear how land surface

hydrology at a global scale may be influenced by El Niño and whether such an influence may
be more region-specific even in tropical areas that are close to the El Niño source region



where impacts are generally expected to be more pronounced (Schubert et al., 2016). This
lack of understanding is reflected in substantial multi-spatial and temporal scale errors in
ENSO impacts on hydrology in models (Zhuo, et al., 2016). Of the land surface hydrologic
variables, soil moisture is of particular interest due to the scarcity of observations available
to properly evaluate its response to El Niño (Gruber et al., 2018), particularly in the low
latitude tropics (Dorigo et al., 2011), as opposed to the more well-studied response of
precipitation over the same region (Ropelewski and Halpert, 1989; Dai and Wigley, 2000;
Chou et al., 2009; Huang and Chen, 2017; Xu 2017).  Moreover, understanding soil moisture
variability to macroclimatic events is useful because of its role in partitioning the energy
fluxes at the Earth's surface (Seneviratne et al., 2010), as well as its importance as a driver
of tropical biomass productivity (Raddatz et al., 2007) and ecosystem responses within Earth
System Models (ESMs) (Green et al., 2019).

Several additional factors highlighted in previous studies contribute to the

uncertainty of how soil moisture will respond to El Niño for different areas. A study in which
soil moisture anomalies were regressed against the Southern Oscillation Index (SOI), one of
the indices of ENSO intensity, revealed that within the tropics, soil moisture typically
decreases during El Niño events, with notable exceptions occurring in extreme southern
Africa and parts of South America (Miralles et al., 2014). However, much of the data used in
the analysis from the tropics were actually missing because they were derived from active
and passive microwave satellite sensors that fail to penetrate the ground beneath dense
rainforests, resulting in substantial data gaps throughout the tropical regions (Liu et al.,
2012; Dorigo et al., 2017). Another study used a coupled biosphere-hydrology model
simulation and determined that soil moisture decreased in the Amazon basin during the



2015-16 super El Niño with more acute reductions occurring in the northeastern part of the
basin (van Schaik et al., 2018). Given that the study did not assess changes over the region
during other super El Niño events, it is unclear if a similar spatial pattern emerges during El
Niños that are comparable in magnitude.

Building on these previous studies, we evaluate the soil moisture response to El Niño

within the humid tropics from 1979 to 2016 with a focus on three super El Niño events. We
concentrate our assessment on soil moisture because of its strong controls on energy and
water exchanges at the land-atmosphere interface and because it represents the main source
of water for natural and cultivated vegetation (Prigent et al., 2005). Soil moisture data for
the analysis was derived from the monthly Global Land Data Assimilation System (GLDAS)
products at one-degree resolution, which are spatially continuous across the globe since
January 1979 (Rodell et al., 2004). The continuous temporal resolution of this data product
satisfies one of our goals by enabling evaluation of the soil moisture response across the
three super El Niños: 1982-83, 1997-98 and 2015-16, which has never before been done.
The continuous spatial coverage of GLDAS enables analysis of the soil moisture response
across all tropical regions, including dense rainforests, which was limited to less densely
forested areas in studies reliant on remote sensing (e.g. Miralles et al., 2014).

**Methods**

GLDAS data was downloaded from the Giovanni online data system, which is

maintained by the National Aeronautics and Space Administration Goddard Earth Sciences
Data and Information Services Center (NASA GES DISC, Acker and Leptoukh, 2007). Data
from GLDAS is derived from precipitation gauge records, satellite data, radar precipitation





observations and various outputs from numerical models (Rodell et al., 2004). We used
1979-2016 monthly data from all four GLDAS land surface models (LSMs) including the
Variable Infiltration Capacity (VIC) model (Liang et al., 1994), Community Land Model (CLM)
(Dai et al., 2003), Noah LSM (NOAH) (Ek et al., 2003) and the Mosaic LSM (MOSAIC) (Koster
and Suarez, 1996). GLDAS soil moisture data is used as the basis for this analysis because
soil moisture estimates from the four individual GLDAS LSMs capture the range of variability
in other similar global soil moisture data products at the locations of the in-situ data that was
used in this study and described in Table 1 (Fig. 1). Other data products in this comparison
include the fifth generation European Center for Medium-Range Weather Forecasts
(ECMWF) reanalysis soil moisture product (ERA5) (Copernicus Climate Change Service
(C3S), 2017), the Modern-Era Retrospective analysis for Research and Applications, Version
2 (MERRA2) (Gelaro et al., 2017) and the Global Land Evaporation Amsterdam Model
(GLEAM) (Miralles et al., 2011; Martens et al., 2017). All three datasets have a spatial
resolution of 0.25°. To avoid integration of results from different climate zones, which are
likely to show a dissimilar soil moisture response, we targeted only GLDAS pixels considered
to be part of the humid tropics by creating a mask using data from the Köppen-Geiger climate
classification system (Kottek et al., 2006) obtained from the Spatial Data Access Tool (SDAT)
(ORNL DAAC, 2017a). The mask was used in conjunction with the monthly soil moisture
estimates to isolate changes specific to the tropical climate zone.
In addition to the four data products mentioned above, we also considered using the
European Space Agency Climate Change Initiative (ESA CCI) global soil moisture product
(Dorigo et al., 2017). However, because this product is derived from observations from
satellite microwave sensors that have difficulty retrieving data beneath dense rainforest



canopies, ESA CCI soil moisture estimates within the tropics were too sparse to reliably
determine the spatially continuous soil moisture response to El Niño across all tropical
regions (e.g. Liu et al., 2012).

Soil moisture is represented in each of the four GLDAS LSMs in a sequence of

subsurface layers up to a maximum of three to ten layers. Each subsurface layer represented
in GLDAS varies in depth up to an aggregated, multi-layer maximum depth of 3.5 m among
the four models.  We only used data from the uppermost group of soil layers within each
model closest to a depth of 0-10 cm. This was done to target the near-surface soil moisture
response to El Niño, as the El Niño signature in soil moisture at shallow depths is likely to be
more prominent and the largest number of in-situ observations that are available for
comparison to the GLDAS estimates also come from the near surface zone. We used the
ensemble mean at 0-10 cm depth from the four models because the ensemble is considered
to provide a more robust representation of reality (Tebaldi and Knutti, 2007).

Soil moisture estimates from GLDAS were validated through comparison to in-situ

observations across 16 sites spanning five continents (Table 1). These data were accessed
through a variety of sources including the Cosmic-ray Soil Moisture Observing System
(COSMOS) (Köhli et al., 2015), United States Department of Agriculture Soil Climate Analysis
Network (SCAN) (Schaefer et al., 2007), Plate Boundary Observatory (PBO) (Larson et al.,
2008), International Soil Moisture Network (ISMN) (Dorigo et al., 2011; Dorigo et al., 2013),
several FLUXNET sites (Goulden et al., 2004; Beringer et al., 2007; Bonal et al., 2008; Beringer
et al., 2011; Beringer et al., 2013) and other individual data collaborators who have made
their data available for use in this study. Data from the individual GLDAS LSMs were
interpolated to the same depths as the in-situ data shown in Table 1 using cubic spline and



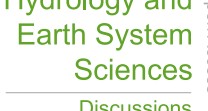

linear interpolation prior to ensemble averaging and comparison with the in-situ data. When
interpolating data from CLM, which includes soil moisture estimates for ten distinct
subsurface layers, cubic spline interpolation was used. Linear interpolation was used for the
other three GLDAS models, which include soil moisture estimates from either three or four
distinct subsurface layers where cubic spline interpolation would have been less
appropriate. The GLDAS data was compared to in-situ data using the linear relationship
shown in Equation 1:

$$SM_I = \beta_0 + \beta_1 * SM_G \qquad\qquad\qquad (1)$$


where $SM_I$ is the in-situ soil moisture observation (%), $\beta_0$ is the y-intercept (%), $\beta_1$ is the
slope and $SM_G$ is the GLDAS ensemble soil moisture estimate (%). The coefficients of the
linear relationship in Equation 1 were used to provide a bias-corrected estimate of soil
moisture from GLDAS that was more representative of the near-surface in-situ soil moisture
observations. The bias-corrected estimates are compared to in-situ observations to assess
how application of the bias-correction method improves the representation of soil moisture
at the point scale.

In-situ soil moisture observations were compared to corresponding GLDAS soil

moisture estimates at co-located depths for pixels that encompassed the in-situ observation.
In some situations, adjacent pixels were used if data from the co-located GLDAS pixel was
missing, e.g., over lands adjacent to inland water bodies or oceans, due to the coarse
resolution of the GLDAS dataset. The same data comparison was made after removing data
from one site in Ecuador and another from Australia. In-situ observations from these sites



were not likely to be representative of the GLDAS data at one-degree resolution given that
the sites where data was collected are either located at a high elevation of 3,780 m or
seasonally flooded wetland where the sub-surface soil is frequently saturated. Observations
from one site in Brazil were also removed due to poor agreement between observations and
GLDAS data relative to other sites.

Comparison of soil moisture from GLDAS to in-situ point-based measurements does

have an inherent scale mismatch. For example, measurements at an individual site may not
necessarily represent soil moisture conditions at the scale of a GLDAS pixel due to
heterogeneities in land cover, soil or topography. However, given the previously noted
challenges regarding the dearth of large-scale moisture measurements in the tropics, the
site-based data represent the best available source of actual soil moisture contents in this
region. Scale mismatch effects are also moderated by use of multiple sites spanning multiple
continents. Site-based measurements of soil moisture considered to be outliers in terms of
how they compare to the co-located GLDAS pixel soil moisture estimate are examined further
in the discussion section.

The soil moisture response to El Niño for the three super El Niño events of 1982-83,

1997-98 and 2015-16 was calculated by taking the difference in the GLDAS soil moisture
during the El Niño mature phase of October to December (OND) and January to March (JFM)
from the long-term 1979-2016 climatological monthly mean (Eqs. 2 and 3):

$\Delta SM_{OND} = SM_{OND} - \sum_1^n SM * n^{-1}$ (2)

$\Delta SM_{JFM} = SM_{JFM} - \sum_1^n SM * n^{-1}$ (3)






where SM is the 3-month mean GLDAS soil moisture during the mature phase (either OND
or JFM) of the focal year for three super El Niños (1982-83, 1997-98 and 2015-16) and n
indicates the total number of monthly estimates used in the analysis from 1979-2016.

K-means cluster analysis was used to determine groups of pixels representing soil

moisture anomaly with a similar magnitude and direction of change during OND and JFM
across the three super El Niño events. Clustering was based on the $\Delta$SM for OND and JFM
that were calculated using Equations 2 and 3. Prior to conducting the analysis, the $\Delta$SM
values were re-scaled to have a mean of 0 and standard deviation of 1. The mean and
standard deviation of OND and JFM $\Delta$SM within each clustered region was then used to
assess the consistency of soil moisture response for different clustered regions.

The number of clusters used in the K-means cluster analysis was set to four. This

number was selected based on results from a suite of tests used to determine the optimal
number of clusters using the R package NbClust (version 3.0) (Charrad et al., 2014). Each
test uses a set of criteria to generate a score for the proposed number of clusters (ranged
between four and ten). We used only tests where the optimal number of clusters was based
on which proposed number of clusters had the maximum or minimum score so the proposed
cluster groups could be ranked accordingly. The mean ranking for all tests across all periods
(OND and JFM for three super El Niños) was then used to determine the optimal number of
clusters (Table 2).

The response of precipitation and evapotranspiration (also obtained from GLDAS) to

El Niño was also determined to compare against the soil moisture responses. The
precipitation and evapotranspiration responses ($\Delta$P and $\Delta$ET) to the three super El Niños
are calculated following the same metric for the soil moisture responses ($\Delta$SM) shown in





Equations 2 & 3. The OND and JFM ΔSM is compared to ΔP and ΔET for the three super El
Niños and plotted on maps as the ΔSM:ΔP and ΔSM:ΔET ratios. The pixel-wide mean ΔSM:ΔP
and ΔSM:ΔET ratios and standard deviations for each of the four clustered regions during
OND and JFM are also summarized.

The relationship between soil moisture and El Niño is further evaluated by calculating

the Pearson correlation coefficient (r) between the 1979-2016 GLDAS monthly soil moisture
and the monthly Niño-3.4 index (Trenberth, 1997; Bunge and Clarke, 2009) for all GLDAS
pixels in the humid tropics. The Niño-3.4 index is a variant of the Niño-3 index region in that
it is centered further west (120 – 170° W vs 90 – 150° W) at the same latitude range (5° N –
5° S). The Niño-3.4 index data was downloaded from the NOAA/OAR/ESRL PSD, Boulder,
Colorado web site at http://www.esrl.noaa.gov/psd (accessed 24 October 2017). The mean
correlation was calculated and summarized for the same regions that were derived from the
cluster analysis. The same correlation analysis was conducted using the soil moisture
response lagged by up to six months for the four clustered regions during OND and JFM.
Because this failed to increase the amount of variability in soil moisture estimates that could
be explained by Niño-3.4 over any of the clustered regions by more than 1%, we only present
correlation results with no lag.

Finally, we calculated the soil moisture response to El Niño for the tropics using the

bias-corrected estimates of GLDAS soil moisture that were derived from the comparisons
with the in-situ soil moisture data. We compare this to the unbiased estimates to determine
the spatial variability in the magnitude of mismatch between these two estimates. Given the
limited number of in-situ observations that were available to generate the bias-corrected
estimates, we use this only to highlight regions where a higher density of soil moisture

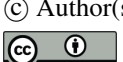



observations might be necessary to improve the accuracy of the soil moisture response to El
Niño derived from GLDAS.

**Results**

GLDAS soil moisture estimates were validated against all in-situ soil moisture

estimates as well as through the removal of three outliers (Fig. 2). Exclusion of the Ecuador,
Australia and Brazil data resulted in an overall reduction in the number of observations by
15% but dramatic improvement in the $r^2$ between GLDAS and in-situ estimates from 0.03 to
0.54. Comparison of these datasets following the removal of outliers reveals a consistent
positive bias in the GLDAS soil moisture estimates relative to in-situ observations.
Consequently, the equation from the best-fit linear regression line (Eq. 1) was used to reduce
the bias in the GLDAS estimates (Fig. 2). Use of the bias-corrected soil moisture estimates
from GLDAS resulted in a mean reduction of RMSE across all sites by 69% (Fig. 3). The
resulting RMSE and $r^2$ coefficient of determination across these sites ranged from 0.03-0.24
(mean = 0.08) and 0.00 to 0.88 (mean = 0.45), respectively (Fig. 4).

Our results of soil moisture changes over regions derived from the cluster analysis

show that the most consistent and strongest decreases in OND soil moisture during the three
super El Niño events occurred over the northeast Amazon Basin and maritime southeast Asia
(Fig. 5a). Regions with the largest and most consistent increases in OND soil moisture over
the three events include eastern and southern equatorial Africa, Latin America and southern
India. Notably, the positive anomalies are much larger during 1982 and 1997 than in 2016.
During the late mature phase of El Niño (JFM), the strongest and most consistent decreases
in soil moisture during the three super El Niño events were centered north of the equator,





while consistent increases largely occurred south of the equator (Fig. 5b). This pattern holds
more or less consistent across the three major land masses of South America, Africa and
Asia/Australia. The largest overall increase in soil moisture was centered over the southern
Amazon Basin. Similar to the changes observed during OND, the positive anomalies tended
to be larger during the two earlier El Niños of 1983 and 1998.

Four clusters are shown for each of the OND (Fig. 6a) and JFM (Fig 6b) periods. The

cluster with the highest soil moisture increases is Cluster-3 followed by Cluster-4, while the
highest soil moisture decreases are found in Cluster-2 followed by Cluster-1. The cluster
results during OND confirm the locations of the largest, most consist soil moisture decreases
(denoted by Cluster-2) over the northeast Amazon Basin and increases (denoted by Cluster-
3) over east Africa, Latin America and southern India (Fig. 6a). The mean decrease in soil
moisture over the Cluster-2 region during OND varied between -0.06 to -0.15 over the three
super El Niño events, while the mean increase in soil moisture over the Cluster-3 region
varied between 0.04 to 0.06 (Table 3). Similarly, during JFM the cluster results show
decreases centered north of the equator and increases south of the equator with smaller
overall coverage of Cluster-3 occurring in 2016 (Fig. 6b). The cluster results during JFM
confirm the locations of the largest, most consistent soil moisture decreases (denoted by
Cluster-2) over the northeast Amazon Basin and increases (denoted by Cluster-3) over east
Africa, Latin America and southern India (Fig. 6b). The mean decrease in the Cluster-2 region
during JFM varied between -0.10 to -0.12 over the three super El Niño events, while the mean
increase in Cluster-3 varied between 0.08 to 0.12 (Table 3).

The change in soil moisture during El Niño is generally tracking that of precipitation

based on the ratio of $\Delta$SM to $\Delta$P. Both $\Delta$SM to $\Delta$P were normalized by their respective 1979



to 2016 mean value prior to calculating the ratio (Fig. 7a and Fig. 7b). Major exceptions to
precipitation tracking soil moisture occurred in the Cluster-4 region where the mean
direction of change in precipitation was opposite that of soil moisture during OND 1982 and
2015, as well as JFM 1998 (Table 4). Moreover, the mean magnitude of soil moisture change
was greater than 12 times that of precipitation, which was at least 2.5 times larger than the
magnitude difference reported for other regions. Much of these anomalies are attributed to
the lack of agreement between precipitation and soil moisture direction of change occurring
in the southern Amazon Basin, Latin America and equatorial Africa including the Sahel. An
amplified soil moisture response, particularly in the Sahel during OND 1997 and the
southern Amazon Basin during OND 1997 and 2015, may be an indication of an important
role of land-atmosphere interactions and/or temperature effects.

Similarly, the deviation of soil moisture is in general tracking that of

evapotranspiration based on the ratio of ΔSM to ΔET (Fig. 8a and Fig. 8b). Many of the same
exceptions to this pattern that were noted with precipitation were also observed here—the
mean direction of change in evapotranspiration was opposite to that of soil moisture
primarily in the Cluster-4 region during OND and JFM 1997-1998, as well as JFM 1983 (Table
5). The lack of agreement in the direction of evapotranspiration and soil moisture change is
also strongest in the southern Amazon Basin, Latin America and equatorial Africa including
the Sahel, particularly during OND 1997 and JFM 1998. Amplification of soil moisture
relative to evapotranspiration also occurred, especially in the southern Amazon Basin and
equatorial Africa during OND 1997 and JFM 1998.

The Pearson correlation coefficient (r) between GLDAS soil moisture and the Niño-

3.4 index for the humid tropics across the 38-year record is provided in Figure 9. In most



regions, there is an inverse relationship indicating the occurrence of El Niño leads to
decreased soil moisture within the tropics. The mean correlation over the clustered regions
are provided in Table 5, which indicates that the strongest mean negative correlations of -
0.12 and -0.09 occurred in Cluster-2 during OND and JFM, respectively. The Cluster-2 group
includes the Amazon Basin, Sahel, southeast Asia and maritime southeast Asia, many of
which were also shown to have the strongest and most consistent decreases in soil moisture
during the super El Niños. The strongest positive correlation of 0.05 occurred in Cluster-3
during JFM, which includes the southern Amazon Basin, east Africa and northern Australia.
These same regions also had the strongest and most consistent increases in soil moisture
during the super El Niños.

Figures 10a and 10b show the difference in OND and JFM soil moisture anomalies

with the addition of the bias correction that was developed using the in-situ data (e.g. Eq. 1
and Figure 1). For both OND and JFM, the application of the bias-corrected estimate
effectively led to a strengthening of the change in soil moisture anomalies relative to the
original GLDAS estimates. The strengthening of the magnitude generally falls between -0.05
and +0.05 with higher values occurring in regions where the original change in soil moisture
anomaly magnitude is higher in Figures 5a and 5b, such as the northeast Amazon Basin and
east Africa.

**Discussion**

Our findings generally agree with Miralles et al., (2014) who also reported a decrease

in soil moisture over the eastern Amazon Basin, Sahel, mainland southeast Asia and northern
Australia, as well as an increase over east Africa. Similar to van Schaik et al., (2018), we found





more acute reductions in soil moisture over the northeastern part of the Amazon Basin
during OND, but the center of these reductions shifted further west during JFM. This is shown
in Figures 5a and 5b as well as Cluster-2 in Figures 6a and 6b, which indicates the decrease
in soil moisture anomaly reached a maximum of 0.24 over the Cluster-2 region. However,
our methods allowed for a spatially continuous estimate across regions as well as an
assessment of soil moisture across seasons (e.g. OND vs. JFM), while focusing on super El
Niño events. As a result, we found several key differences in the soil moisture response to El
Niño relative to previous studies. Specifically, this includes increases in the soil moisture
anomaly of up to 0.20 over Latin America during OND, decreases in the soil moisture
anomaly of up to 0.24 over the Sahel during OND, decreases in the soil moisture anomaly of
up to 0.24 over maritime regions of southeast Asia during both OND and JFM, as well as
increases in the soil moisture anomaly of up to 0.20 over southern India during OND and
northern Australia during JFM.
The southern Amazon Basin stuck out as one region where the direction or magnitude
of change in soil moisture did not necessarily match that of precipitation or
evapotranspiration. This may in part be due to the distinction in climate impacts between
the northern and southern Amazon Basins during an El Niño event. The northern Amazon
Basin is influenced by displacement of the Intertropical Convergence Zone (ITCZ) and
changes in the Hadley cell positioning during this time, which forces the ITCZ northward
resulting in a reduction of rainfall (Marengo, 1992). However, the southern Amazon Basin is
primarily dependent on the South Atlantic Conversion Zone (SACZ), which is not as
influenced by El Niño. In general, during the peak El Niño season the intensification of the
SACZ enhances the southerly flow of low-level jets (LLJs). Circulation blockages produced by





the Andes help to channelize and intensify the LLJs over the southern Amazon Basin,
resulting in LLJs having primary control on temperature and precipitation regimes within
the region during the austral summer. Consequently, the southern Amazon Basin actually
experiences more rain during this time, but predictability of the timing and magnitude of this
sequence events and associated impacts on rainfall is generally lower than that of El Niño for
the northern Amazon (Marengo et al., 2002; Marengo et al., 2004). Moreover, rainfall
processes in the southern Amazon Basin depend on the displacement of cold fronts and
mesoscale circulation patterns, which occur at the synoptic scale. Thus, the lack of agreement
between precipitation and evapotranspiration change with soil moisture change in this
region occurs because of the strong impacts of atmospheric processes that originate outside
of this region (Silva Dias et al., 2002).

The spatial patterns we identified indicate that the relationship between soil

moisture and El Niño is more nuanced than what is revealed from the correlation of soil
moisture with the Niño-3.4 index. Although this analysis still indicates much of South
America, mainland southeast Asia and nearby islands respond most strongly to El Niño, the
pixels with stronger correlations do not precisely align with the regions identified where the
most consistent directional change during the three super El Niño events was observed. For
example, weak correlations ($|r| < 0.2$) between soil moisture and Niño-3.4 were identified
throughout the Sahel, Latin America and mainland southeast Asia during both OND and JFM,
despite portions of these regions showing a consistent positive or negative change in soil
moisture during super El Niño events. Several factors might be contributing to this issue.
First, as shown in Figures 1 and 2, the Sahel shows more widespread decreases in soil
moisture during OND, but increases during JFM. Thus, the inverse weak correlation in this





region might be occurring due to contrasting changes in soil moisture brought on by El Niño
during the first and second halves of the peak El Niño season. Second, we targeted the three
most recent super El Niños to evaluate the tropical soil moisture response, while the Niño-
3.4 index does not distinguish between the magnitude or type (e.g. CP or EP) of El Niño (Kao
and Yu, 2009; Yu and Zou, 2013). As such, the correlations shown in Figure 8 are more
representative of mean El Niño conditions, while the soil moisture changes depicted in
Figures 5a and 5b are representative of super El Niño conditions. We refrained from
conducting the correlation between soil moisture and the Niño-3.4 index using only months
when the three super El Niños occurred because this would severely limit the number of
observations available for use in the analysis. Another potential issue is related to the
accuracy of the GLDAS soil moisture response to El Niño for the tropics, which was dealt with
through comparison to in-situ observations.

Although the bias correction applied to GLDAS soil moisture shown in Figures 2 and

3 were able to substantially reduce the RMSE between in-situ observations and GLDAS
estimates, the overall performance of GLDAS in terms of $r^2$ is still mixed. Ten of the in-situ
sites that were evaluated had an $r^2 > 0.4$, while four had an $r^2 < 0.1$ (Figure 4). The large
disagreement between in-situ data and GLDAS for some locations is likely to be the result of
a mismatch in scale between these two datasets. As a result, GLDAS pixels with greater
topography, land cover or soil heterogeneity are less likely to match in-situ observations. For
instance, in the Manaus region of central Amazon, soils can vary from greater than 90% clay
on plateaus to greater than 90% sand in valleys at a horizontal distance of only 500 m and
the soil moisture can vary from over 100% in this span (Chauvel et al., 1987; Tomasella et
al., 2008; Cuartas et al., 2012). During dry periods such as those that typify a peak super El





Niño event for this region, strong variations in soil moisture have been detected at depths of
up to 5 m (Broedel et al., 2017). Because the maximum soil depth represented by GLDAS is
restricted to more shallow soil layers, the soil moisture variability represented in GLDAS for
this region should be taken with caution. Ideally, multiple in-situ observations at greater soil
depths could be used for comparison to each GLDAS pixel that was tested, but this level of
data coverage is generally not available for soil moisture, particularly in tropical regions
(Brocca et al., 2017). Although GLDAS also includes a 0.25-degree soil moisture product, the
higher spatial resolution data only includes estimates from one model and does not provide
estimates from all three of the most recent super El Niños.

Given the bias observed in the GLDAS soil moisture product relative to in-situ data
over the available record, we also compared soil moisture estimates from GLDAS to in-situ
data only during the mature phase 2015-16 super El Niño event to confirm that a similar bias
occurred during this period. The variability of in-situ estimates captured by GLDAS differed
by no more than 2% when considering only the peak El Niño months of the 2015-16 event,
thereby demonstrating that the variability in bias between the two periods was minimal.
Given the higher number of observations when all months were used (e.g. n= 802 versus only
n= 67), we chose to base the bias-corrected estimate on the comparison made using all
available months of data to incorporate a greater number of observations into the analysis.

Several strategies exist that can increase confidence in soil moisture estimates from
data products like GLDAS. First, in-situ observations of soil moisture need to improve in both
space and time to evaluate and constrain the land surface models used in GLDAS. The
distribution of soil moisture observations is much lower in tropical regions than other areas
(Brocca et al., 2017), which is not surprising given the dearth of hydrologic observations





available from developing countries in tropical regions (Alsdorf et al., 2007) coupled with
the reported decrease in hydrologic monitoring across sites worldwide (McCabe et al.,
2017). In addition, increased participation in contributing in-situ soil moisture data to online
databases such as FLUXNET (ORNL DAAC, 2017b) and ISMN (Dorigo et al., 2011; Dorigo et
al., 2013) would help alleviate the limited access to observational datasets.

Satellite observations of soil moisture can also be used to fill this gap, but a number

of issues exist with historical satellite derived estimates of soil moisture. Substantial biases
exist in retrieval algorithms (Entekhabi et al., 2010) and direct estimates are restricted to
shallow soil depths are of limited value when soil moisture at greater depths is needed
(McCabe et al., 2017). Such shortcomings have encouraged investigations into the relative
influence of vegetation, soil and topography on soil moisture dynamics to better upscale
point-based measurements of soil moisture to larger, remotely sensed scales (Gaur and
Mahanty, 2016). Algorithms have been developed to interpolate shallow subsurface
estimates of soil moisture to the root zone, but a recent global evaluation of the accuracy of
the algorithms being used for this purpose to generate Soil Moisture Active Passive (SMAP)
Level 4 data was limited to 17 sites with only one occurring within the tropical climate zone
(Reichle et al., 2017). Moreover, satellite radar used to observe soil moisture from many
historical missions fails to penetrate dense rainforest canopies making this data of limited
use for many tropical regions. Another issue with satellites is the limited lifetime of the
mission coupled with the lack of follow-on missions that would enable extension of the
observation record so impacts from cyclical climate events like ENSO that occur on decadal
timescales can be adequately assessed. As a result, data is often combined from multiple
missions to extend satellite records, which can introduce additional error (Gruber et al.,



2019). Access to more spatially and temporally continuous global soil moisture data from
satellites or assimilation products are thus paramount to improve the spatial and temporal
resolution of soil moisture estimates and enable better prediction of soil moisture behavior
over long timescales (Brocca et al., 2017).

Lastly, the current GLDAS product is produced mainly by running offline land surface

models forced with atmospheric data from a combination of rain gauge, satellite, and radar
precipitation estimates and outputs (e.g., radiation) from numerical prediction models.
Uncertainties and biases in the land models and forcing data can contribute importantly to
uncertainties and biases in the GLDAS soil moisture (Piao et al., 2013). Future products that
assimilate in-situ and remotely-sensed observations of terrestrial energy and water storages
such as soil moisture and snow and fluxes such as evapotranspiration, sensible heat flux, and
runoff will likely further improve the quality of GLDAS soil moisture for better
characterization of impacts from El Niño (e.g. Albergel et al., 2012; Gruber et al., 2018). This
has important implications for understanding water resources and plant response to ENSO
events, given the role of soil moisture in climate extremes due to feedbacks with the
atmosphere (Seneviratne et al., 2010).

**Summary and Conclusion**

We describe the response of soil moisture in the humid tropics to El Niño while

focusing on impacts from the three most recent super El Niños of 1982-83, 1997-98 and
2015-16 using soil moisture estimates from GLDAS. The largest and most consistent
reductions in the soil moisture anomaly of up to 0.24 occurred over the northern Amazon
basin and the maritime regions of southeast Asia, Indonesia and New Guinea. The soil





moisture response is largely consistent with the precipitation and evapotranspiration
responses, as indicated by the overwhelmingly positive ratio of soil moisture change to both
precipitation and evapotranspiration change over the same period in regions with consistent
soil moisture response. Some notable exceptions include the Sahel and southern Amazon
Basin where a greater number of pixels show the direction of change for soil moisture is
opposite that of precipitation and evapotranspiration. The soil moisture change was
amplified relative to precipitation and evapotranspiration in these areas particularly during
OND, suggesting that the soil moisture response may be amplified through land-atmosphere
interactions and/or the temperature response and differing climate patterns between the
north and south Amazon Basin. Indeed, land-atmosphere interactions have been suggested
to play more of an important role in the regional water cycle over the Amazon and Sahel (e.g.,
Koster et al. 2004; Wang et al., 2013; Levine et al., 2019), so their role in the soil moisture
response to El Niño deserves more investigation over these regions in the future.

Comparison of GLDAS estimates to in-situ data from 16 reference sites to gauge the

utility of these estimates in large scale models reveals a high degree of variability in the
performance of GLDAS among the different sites. Although some of the poor performance
can invariably be explained by a mismatch in the scale of in-situ observations to the coarse,
1-degree resolution of GLDAS, improvements in the availability of ground-based soil
moisture observations and access to more data from temporally-continuous, global soil
moisture observing satellite missions that allow for estimates beneath dense rain forest
canopies are necessary to improve upon these estimates by constraining land model
estimates through data assimilation. Such an effort will be useful to increase the accuracy of



tropics hydrology and ecosystem models to make better predictions of El Niño impacts on
land surface hydrology.

**Acknowledgements**

This project was supported as part of the Next Generation Ecosystem Experiments-

Tropics, funded by the United States Department of Energy, Office of Science, Office of
Biological and Environmental Research through the Terrestrial Ecosystem Science Program.
Data obtained from French Guiana were recorded thanks to an "investissement d'avenir"
grant from the Agence Nationale de la Recherche (CEBA, ref ANR-10-LABX-25-01; ARBRE,
ref. ANR-11-LABX-0002-01). Data obtained from one of the Panama sites were recorded
thanks to an award from the National Science Foundation (NSF 1360305). Global soil
moisture data products used in this study are publicly available at the following locations:

• GLDAS: https://disc.gsfc.nasa.gov/datasets?page=1&keywords=GLDAS

• ERA5: https://cds.climate.copernicus.eu/#!/home

• MERRA2: https://gmao.gsfc.nasa.gov/reanalysis/MERRA-2/data_access/

• GLEAM: https://www.gleam.eu/#downloads

In-situ data collected from Indonesia was made possible by the Deutsche
Forschungsgemeinschaft (DFG, German Research Foundation) – project number 192626868
– SFB 990 and the Ministry of Research, Technology and Higher Education (Ristekdikti) in
the framework of the collaborative German – Indonesian research project CRC990. We
would also like to thank Charu Varadharajan for providing valuable assistance with database
access and guidance on data storage during the course of this research. The Pacific Northwest





National Laboratory is operated for the Department of Energy by Battelle Memorial Institute under
contract DE-AC05-76RL01830.

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

**Tables**
**Table 1**: Information on geospatial location, record length and monitoring instruments used
for in-situ observations that were used in the analysis.





**Table 2**: Mean ranking of proposed cluster groups across OND and JFM during three super
El Niños for tests used in R package NbClust (version 3.0). Low scores denote highest
ranking.

**Table 3**: Mean and standard deviation of October to December (OND) and January to March
(JFM) change in the GLDAS soil moisture for clustered regions in the humid tropics. Statistics
computed using OND and JFM GLDAS soil moisture anomalies during El Niño years 1982-83,
1997-98, and 2015-16 relative to the 1979-2016 mean.

**Table 4**: Mean and standard deviation of October to December (OND) and January to
March (JFM) change in soil moisture to precipitation ratio for the same regions shown
in Table 3. Statistics computed using OND and JFM GLDAS soil moisture anomalies
during El Niño years 1982-83, 1997-98, and 2015-16 relative to the 1979-2016 mean.

**Table 5**: Mean and standard deviation of October to December (OND) and January to March
(JFM) change in soil moisture to evapotranspiration ratio for the same regions shown in
Table 3. Statistics computed using OND and JFM GLDAS soil moisture anomalies during El
Niño years 1982-83, 1997-98, and 2015-16 relative to the 1979-2016 mean.

**Table 6**: Mean and standard deviation of 1979-2016 GLDAS soil moisture correlation with
the Niño3.4 index for the same regions shown in Table 3.






**Figures**
**Figure 1**: 1979-2017 monthly time series of mean soil moisture across all in-situ data
locations shown in Table 1 for multiple data products including the GLDAS multi-model
mean (black, solid), MERRA2 (red, solid), ERA5 (blue, solid), and GLEAM (green, solid), as
well as the individual land surface models that make up GLDAS NOAH (black, short dash),
MOSAIC (black, dot), VIC (black, dash dot) and CLM (black, long dash). Note that the GLEAM
time series starts from 1980.

**Figure 2**: In-situ soil moisture vs. GLDAS soil moisture during October to December (OND)
and January to March (JFM) for El Niño years 1982-83, 1997-98, and 2015-16. Each circle
corresponds to one in-situ data point in space and time. The left panel includes data from all
18 sites shown in Table 1, with data from Australia, Ecuador, and Brazil highlighted in blue,
red, and green, respectively. The right panel shows the same information with the Ecuador,
Australia, and Brazil site data removed.  The blue dashed line and red solid line represent
the 1:1 line and the regression line, respectively.

**Figure 3**:  Bias-corrected soil moisture estimates from GLDAS relative to in-situ soil
moisture observations for all sites with the mean RMSE shown in red.

**Figure 4**: Bias-corrected estimate from GLDAS (black line) and in-situ observation (red line)
of soil water content for 16 individual locations in the humid tropics. RMSE and $r^2$ coefficient
of determination for each location are also shown.



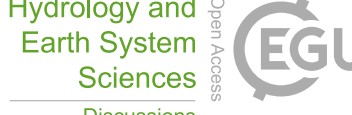

**Figure 5a**: October to December (OND) change in GLDAS soil moisture anomalies during the
super El Niño years 1982 (top), 1997 (middle), and 2015 (bottom) relative to the previous
years. Anomalies relative to 1979-2016 period.

**Figure 5b**: Same as Figure 4a, but for January to March (JFM) in 1983 (top), 1998 (middle)
and 2016 (bottom).

**Figure 6a**: K-means cluster analysis results for October to December (OND) for 1982 (top),
1997 (middle) and 2015 (bottom). Corresponding histograms of soil moisture anomalies for
each of the four clusters also shown. Anomalies relative to 1979-2016 period.

**Figure 6b**: Same as Figure 5a, but for January to March (JFM) in 1983 (top), 1998 (middle)
and 2016 (bottom).

**Figure 7a**: Ratio of GLDAS soil moisture to precipitation change computed using October to
December (OND) anomalies during El Niño years 1982-83, 1997-98, and 2015-16 relative to
previous years. Anomalies relative to 1979-2016 period.

**Figure 7b**: Same as Figure 6a but for January to March in 1983 (top), 1998 (middle) and
2016 (bottom).



**Figure 8a**: Ratio of GLDAS soil moisture to evapotranspiration change computed using October to December (OND) anomalies during El Niño years 1982-83, 1997-98, and 2015-16 relative to previous years. Anomalies relative to 1979-2016 period.

**Figure 8b**: Same as Figure 7a but for January to March in 1983 (top), 1998 (middle) and 2016 (bottom).

**Figure 9**: Pearson correlation coefficient between GLDAS soil moisture and NINO3.4 index from 1979 to 2016. Colors indicate regions where the mean correlation was negative (red) and positive (blue).

**Figure 10a**: Difference in October to December (OND) change in GLDAS soil moisture anomalies when bias correction is applied relative to no bias correction. Red pixels indicate regions that showed a negative correlation increase, while blue regions indicate regions that showed a positive correlation increase with the addition of the bias correction. Plots shown for the super El Niño years 1982 (top), 1997 (middle), and 2015 (bottom).

**Figure 10b**: Same as Figure 9a but for January to March (JFM) in 1983 (top), 1998 (middle) and 2016 (bottom).





**Table 1**: Information on geospatial location, record length and monitoring instruments used for in-situ observations that were used in the analysis.

| Country | Lat (°N) | Long (°E) | Land Cover Type | Record Length (no. months)[1] | Elev (m) | Depth (cm) | Instrument[2] |
|---|---|---|---|---|---|---|---|
| Australia (1) | -17.12 | 145.63 | Rainforest | May 2014 – Mar 2017 (35) | 715 | 28 | COSMOS[4] |
| Australia (2) | -14.16 | 131.39 | Tropical Savanna | Jun 2011 – Dec 2016 (67) | 7.5 | 38 | COSMOS[4] |
| Australia (3) | -13.08 | 131.12 | Woody Savanna | Nov 2007 – May 2009 (19) | 76 | 0-10 | ECFT[5] |
| Australia (4) | -12.49 | 131.15 | Woody Savanna | Aug 2001 – Dec 2014 (161) | 39 | 0-10 | ECFT[6] |
| Australia (5) | -12.55 | 131.31 | Wetlands | Feb 2006 – Oct 2008 (33) | 4 | 0-10 | ECFT[7] |
| Brazil[3] (1/2) | -2.61 | -60.21 | Evergreen Broadleaf Forest | Sep 2015 – Mar 2016 (14) | 130 | 0-10 | TDR[8] |
| Brazil (3) | -3.02 | -54.97 | Evergreen Broadleaf Forest | Jul 2000 – Feb 2004 (44) | 48 | 0-10 | ECFT[9] |
| Brazil (4) | -2.85 | -54.97 | Evergreen Broadleaf Forest | Dec 2008 – Apr 2016 (47) | 200 | 50 | TDR[10] |
| Dom. Republic (1) | 19.76 | -70.57 | Savanna | Feb 2013 – Aug 2017 (53) | -32 | 0-10 | GPS[11] |
| Dom. Republic (2) | 17.90 | -71.67 | Savanna | Feb 2013 – Dec 2016 (56) | -17 | 0-10 | GPS[11] |
| Ecuador | -3.06 | -79.24 | Wet Páramo | Jan 2011 – Dec 2016 (72) | 3,780 | 0-10 | TDR[12] |
| French Guiana[3] | 5.28 | -52.92 | Evergreen Broadleaf Forest | Jan 2007 – Jan 2017 (133) | 20 | 0-10 | ECFT[13] |
| Indonesia | -1.97 | 102.60 | Grassland | Jun 2013 – Sep 2017 (45) | 48 | 30 | TDR[14] |
| Kenya | 0.28 | 36.87 | Savanna/Grassland | Oct 2011 – May 2017 (68) | 1,824 | 15 | COSMOS[4] |
| Malaysia | 1.94 | 103.38 | Orchard | Dec 2014 – Nov 2015 (12) | 88 | 0-5 | TDR[15] |
| Panama (1) | 9.16 | -79.84 | Evergreen Broadleaf Forest | Jul 2012 – Nov 2017 (65) | 330 | 0-10 | TDR[16] |
| Panama (2) | 9.21 | -79.75 | Evergreen Broadleaf Forest | Jul 2015 – Dec 2017 (30) | 203 | 0-10 | EF[17] |

[1] Data not necessarily temporally continuous for every location
[2] COSMOS = Cosmic Neutron Probe, ECFT = Eddy Covariance Flux Tower, EF = Electromagnetic Field, GPS = Global Positioning System, TDR = Time Domain Reflectometry
[3] Comprised of two sites at these coordinates
[4] Köhli et al., 2015
[5] Beringer et al., 2011
[6] Beringer et al., 2007
[7] Beringer et al., 2013
[8] Jardine et al., 2019
[9] Goulden et al., 2004
[10] Wu et al., 2016
[11] Larson et al., 2008
[12] Ochoa-Sánchez et al., 2018
[13] Bonal et al., 2008; and see Acknowledgements
[14] Meijide et al., 2018; and see Acknowledgements
[15] Kang et al., 2016
[16] Rubio and Detto, 2017
[17] Bretfeld et al., 2018





**Table 2**: Mean ranking of proposed cluster groups across OND and JFM during three super El Niños for tests used in R package NbClust (version 3.0). Low scores denote highest ranking.

| Test | 4 | 5 | 6 | 7 | 8 | 9 | 10 |
|------|------|------|------|------|------|------|------|
| KL[1] | 2.83 | 4.33 | 3.83 | 4.17 | 5.17 | 3.50 | 4.17 |
| CH[2] | 5.00 | 6.17 | 5.33 | 3.50 | 3.50 | 1.83 | 2.67 |
| CCC[3] | 3.33 | 4.33 | 3.67 | 4.33 | 4.50 | 3.83 | 4.00 |
| Cindex[4] | 1.50 | 2.00 | 2.83 | 3.83 | 5.33 | 6.17 | 6.33 |
| DB[5] | 4.33 | 2.00 | 2.83 | 2.83 | 4.33 | 6.17 | 5.50 |
| Silhouette[6] | 2.67 | 4.50 | 5.83 | 4.17 | 4.00 | 2.83 | 3.83 |
| Ratkowsky[7] | 1.00 | 2.00 | 3.00 | 4.00 | 5.00 | 6.00 | 7.00 |
| Ptbiserial[8] | 1.33 | 1.67 | 3.00 | 4.17 | 4.83 | 6.00 | 7.00 |
| McClain[9] | 7.00 | 6.00 | 4.83 | 4.17 | 2.83 | 2.00 | 1.17 |
| Dunn[10] | 3.50 | 4.67 | 2.67 | 3.00 | 4.50 | 3.17 | 4.67 |
| SDindex[11] | 7.00 | 5.33 | 4.33 | 4.00 | 3.50 | 2.83 | 1.00 |
| SDbw[12] | 1.00 | 2.00 | 3.17 | 4.00 | 4.83 | 6.17 | 6.83 |
| **Mean** | **3.38** | **3.75** | **3.78** | **3.85** | **4.36** | **4.21** | **4.51** |

[1] Krzanowski and Lai, 1988
[2] Calinski and Harabasz, 1974
[3] Sarle, 1983
[4] Hubert and Levin, 1976
[5] Davies and Bouldin, 1979
[6] Rousseuuw, 1987
[7] Ratowksy and Lance, 1978
[8] Milligan 1980; Milligan 1981
[9] McClain and Rao, 1975
[10] Dunn, 1974
[11] Halkidi et al., 2000
[12] Halkidi and Vazirgiannis, 2001





**Table 3**: Mean and standard deviation of October to December (OND) and January to March (JFM) change in the GLDAS soil moisture for clustered regions in the humid tropics. Statistics computed using OND and JFM GLDAS soil moisture anomalies during El Niño years 1982-83, 1997-98, and 2015-16 relative to the 1979-2016 mean.

| Region[1] | Season | 1982-83, 1997-98, 2015-16 Mean Change ± Standard Deviation |
|---|---|---|
| Cluster-1 | OND | -0.05 ±0.02, 0.01 ±0.02, 0.07 ±0.02 |
| Cluster-2 | OND | -0.12 ±0.02, -0.06 ±0.02, -0.15 ±0.02 |
| Cluster-3 | OND | 0.06 ±0.02, 0.10 ±0.02, 0.04±0.03 |
| Cluster-4 | OND | 0.01 ±0.02, 0.05 ±0.02, -0.01 ±0.02 |
| Cluster-1 | JFM | -0.07 ±0.02, -0.03 ±0.02, -0.06 ±0.02 |
| Cluster-2 | JFM | -0.12 ±0.02, -0.10 ±0.02, -0.12 ±0.03 |
| Cluster-3 | JFM | 0.09 ±0.02, 0.12 ±0.02, 0.08 ±0.02 |
| Cluster-4 | JFM | 0.01 ±0.02, 0.06 ±0.02, 0.02 ±0.01 |



**Table 4**: Mean and standard deviation of October to December (OND) and January to March (JFM) change in soil moisture to precipitation ratio for the same regions shown in Table 3. Statistics computed using OND and JFM GLDAS soil moisture anomalies during El Niño years 1982-83, 1997-98, and 2015-16 relative to the 1979-2016 mean.

| Region[1] | Season | 1982-83, 1997-98, 2015-16 Mean Change ± Standard Deviation |
|---|---|---|
| Cluster-1 | OND | 0.20 ±6.77, -0.01 ±3.11, 4.72 ±53.07 |
| Cluster-2 | OND | -0.15 ±6.42, 0.47 ±12.16, 1.40 ±0.53 |
| Cluster-3 | OND | 0.01 ±6.38, 3.33 ±47.90, 0.26 ±6.40 |
| Cluster-4 | OND | -0.01 ±6.71, 12.35 ±284.91, -1.62 ±130.82 |
| Cluster-1 | JFM | 1.38 ±4.26, 0.47 ±0.95, 29.67 ±1042.43 |
| Cluster-2 | JFM | 1.10 ±0.20, 0.99 ±0.21, 1.33 ±0.86 |
| Cluster-3 | JFM | 1.18 ±1.28, 1.91 ±1.88, 0.92 ±0.37 |
| Cluster-4 | JFM | 0.64 ±22.22, -2.29 ±79.17, 0.72 ±7.77 |





**Table 5**: Mean and standard deviation of October to December (OND) and January to March (JFM) change in soil moisture to evapotranspiration ratio for the same regions shown in Table 3. Statistics computed using OND and JFM GLDAS soil moisture anomalies during El Niño years 1982-83, 1997-98, and 2015-16 relative to the 1979-2016 mean.

| Region[1] | Season | 1982-83, 1997-98, 2015-16 Mean Change ± Standard Deviation |
|---|---|---|
| Cluster-1 | OND | 0.12 ±72.81, -0.21 ±1.12, 1.98 ±24.52 |
| Cluster-2 | OND | 4.45 ±69.16, 0.47 ±1.86, 0.42 ±5.76 |
| Cluster-3 | OND | -0.21 ±75.51, 0.54 ±23.03, 0.53 ±15.53 |
| Cluster-4 | OND | 0.36 ±54.18, -1.13 ±5.46, 4.88 ±135.52 |
| Cluster-1 | JFM | 0.63 ±7.48, -0.82 ±24.18, 1.82 ±28.88 |
| Cluster-2 | JFM | 0.34 ±16.37, 0.99±7.27, 1.87 ±22.99 |
| Cluster-3 | JFM | 0.72 ±2.86, 20.12 ±398.90, 0.67 ±1.89 |
| Cluster-4 | JFM | -0.74 ±8.60, -5.71 ±133.99, 0.34 ±3.39 |





**Table 6**: Mean and standard deviation of 1979-2016 GLDAS soil moisture correlation with the Niño3.4 index for the same regions shown in Table 3.

| Region[1] | Season | Mean Correlation ± Standard Deviation |
|---|---|---|
| Cluster-1 | OND | -0.07 ±0.10 |
| Cluster-2 | OND | -0.12 ±0.13 |
| Cluster-3 | OND | -0.06 ±0.10 |
| Cluster-4 | OND | -0.06 ±0.10 |
| Cluster-1 | JFM | -0.06 ±0.07 |
| Cluster-2 | JFM | -0.09 ±0.07 |
| Cluster-3 | JFM | 0.05 ±0.06 |
| Cluster-4 | JFM | 0.00 ±0.08 |





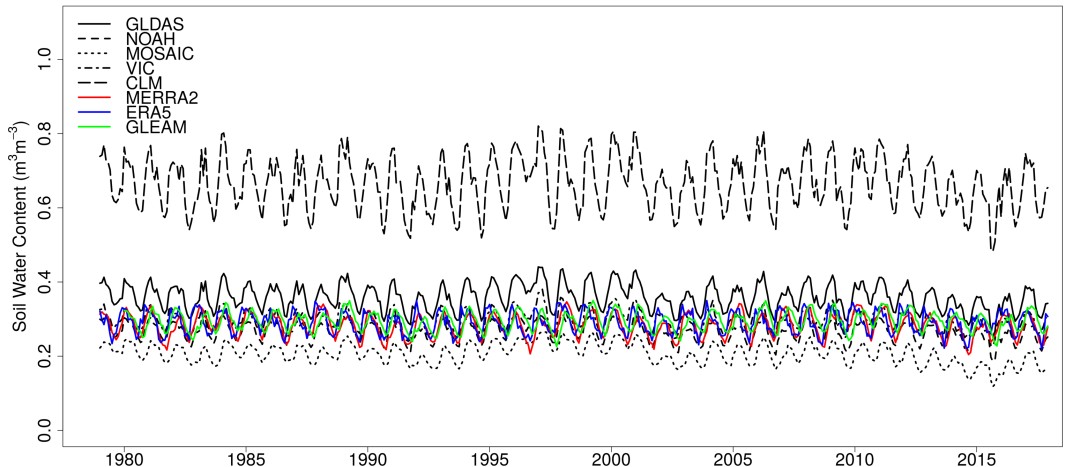

**Figure 1**: 1979-2017 monthly time series of mean soil moisture across all in-situ data locations shown in Table 1 for multiple data products including the GLDAS multi-model mean (black, solid), MERRA2 (red, solid), ERA5 (blue, solid), and GLEAM (green, solid), as well as the individual land surface models that make up GLDAS NOAH (black, short dash), MOSAIC (black, dot), VIC (black, dash dot) and CLM (black, long dash). Note that the GLEAM time series starts from 1980.



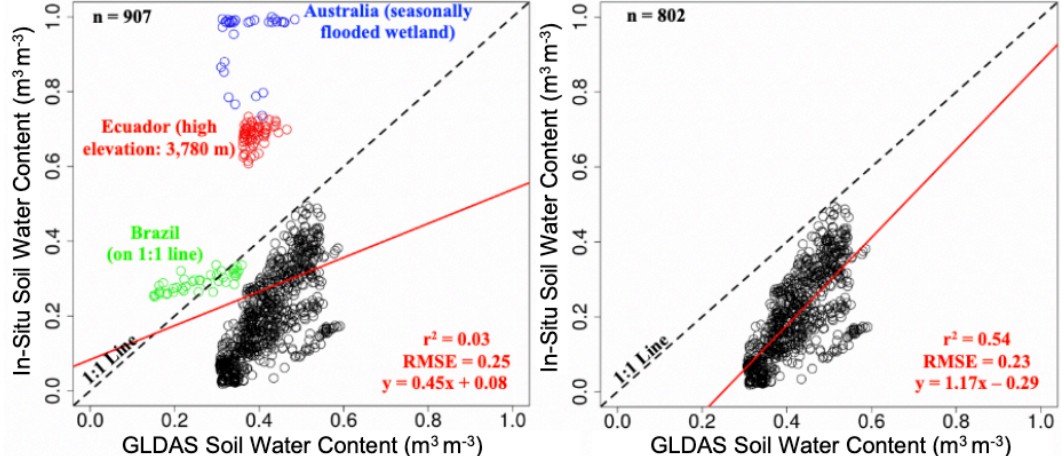

**Figure 2**: In-situ soil moisture vs. GLDAS soil moisture during October to December (OND) and January to March (JFM) for El Niño years 1982-83, 1997-98, and 2015-16. Each circle corresponds to one in-situ data point in space and time. The left panel includes data from all 18 sites shown in Table 1, with data from Australia, Ecuador, and Brazil highlighted in blue, red, and green, respectively. The right panel shows the same information with the Ecuador, Australia, and Brazil site data removed. The blue dashed line and red solid line represent the 1:1 line and the regression line, respectively.





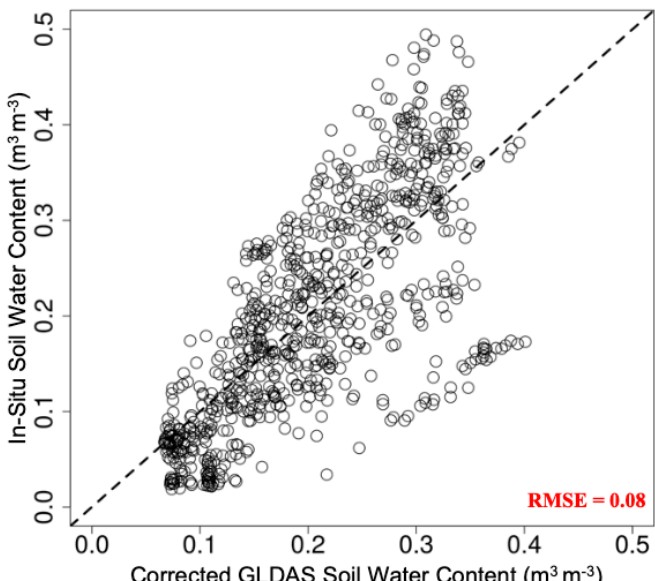

**Figure 3**: Bias-corrected soil moisture estimates from GLDAS relative to in-situ soil moisture observations for all sites with the mean RMSE shown in red.

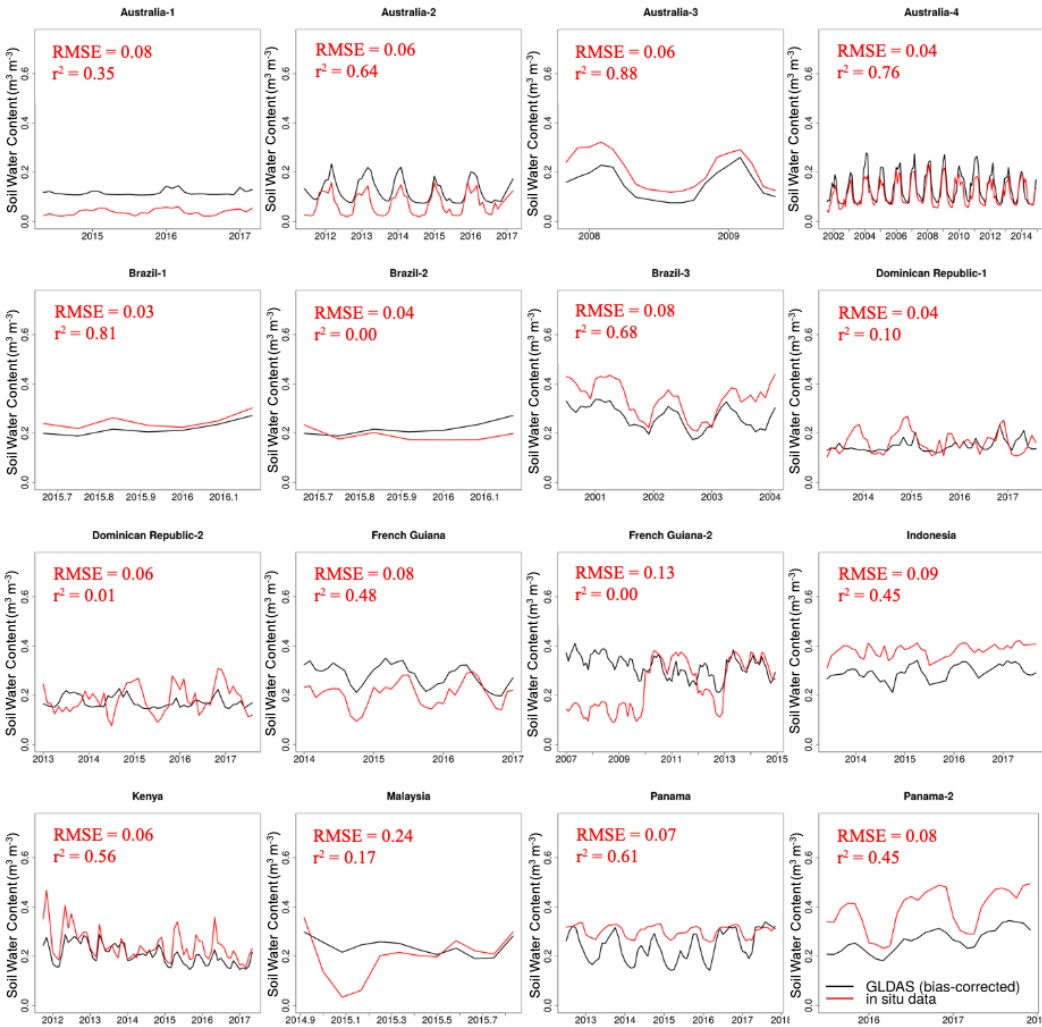

**Figure 4**: Bias-corrected estimate from GLDAS (black line) and in-situ observation (red line) of soil water content for 16 individual locations in the humid tropics. RMSE and r² coefficient of determination for each location are also shown.

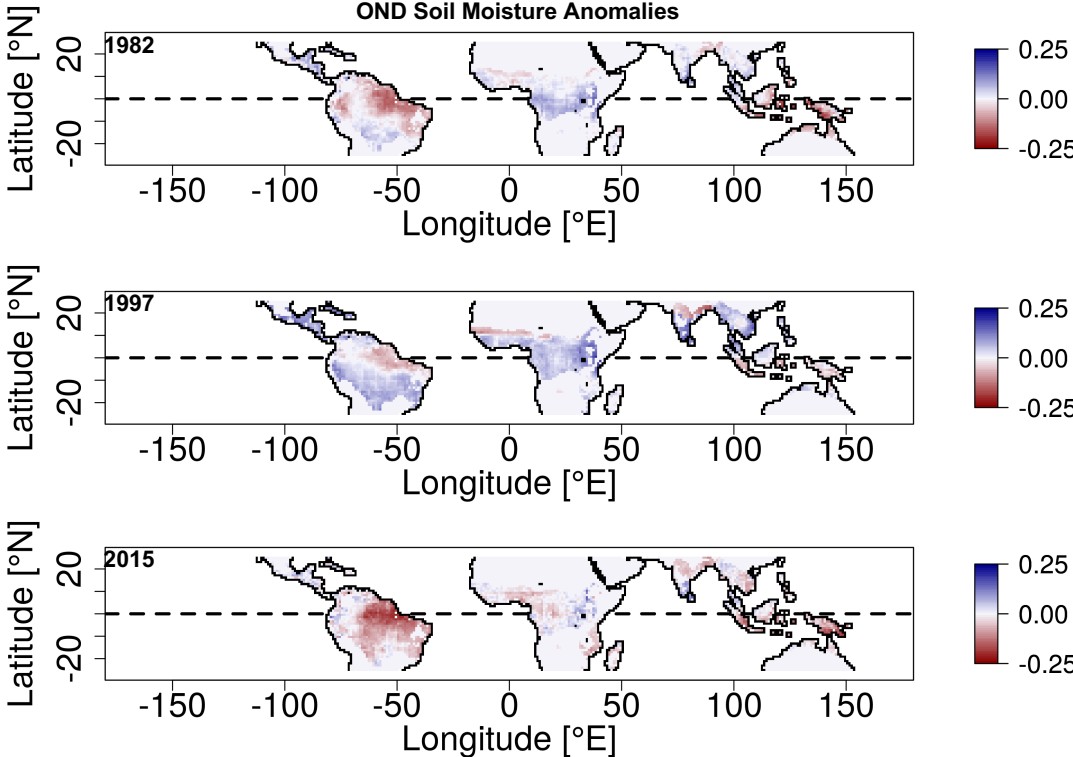

**Figure 5a**: October to December (OND) change in GLDAS soil moisture anomalies during the super El Niño years 1982 (top), 1997 (middle), and 2015 (bottom) relative to the previous years. Anomalies relative to 1979-2016 period.



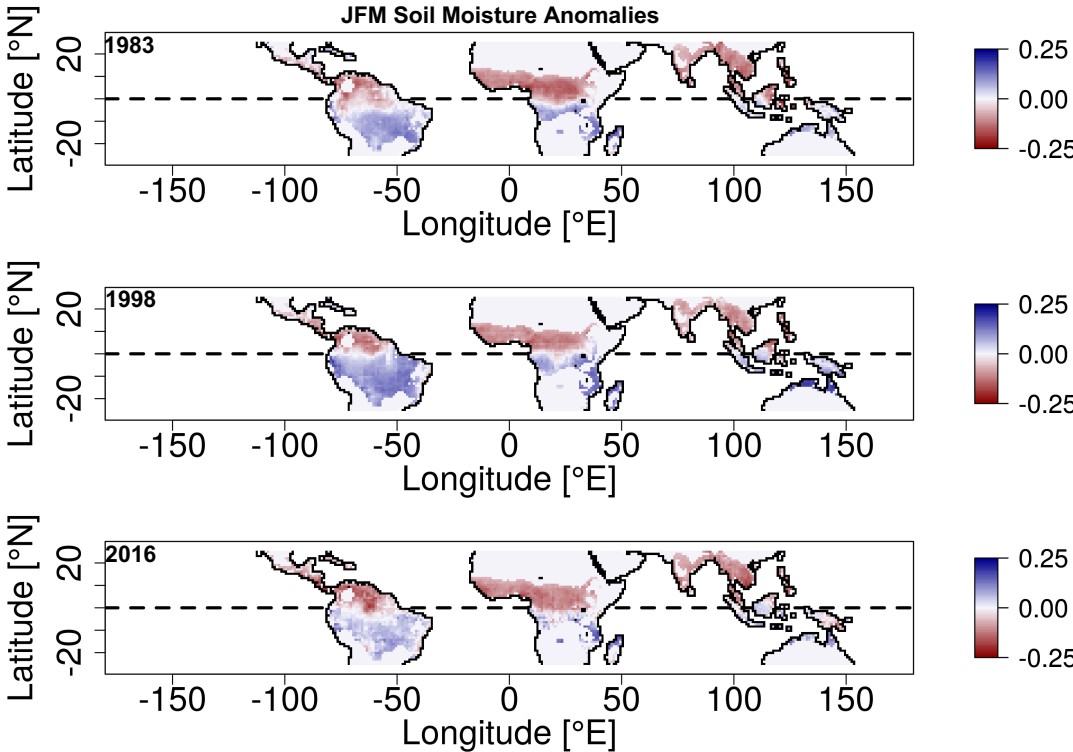

**Figure 5b**: Same as Figure 5a, but for January to March (JFM) in 1983 (top), 1998 (middle) and 2016 (bottom).





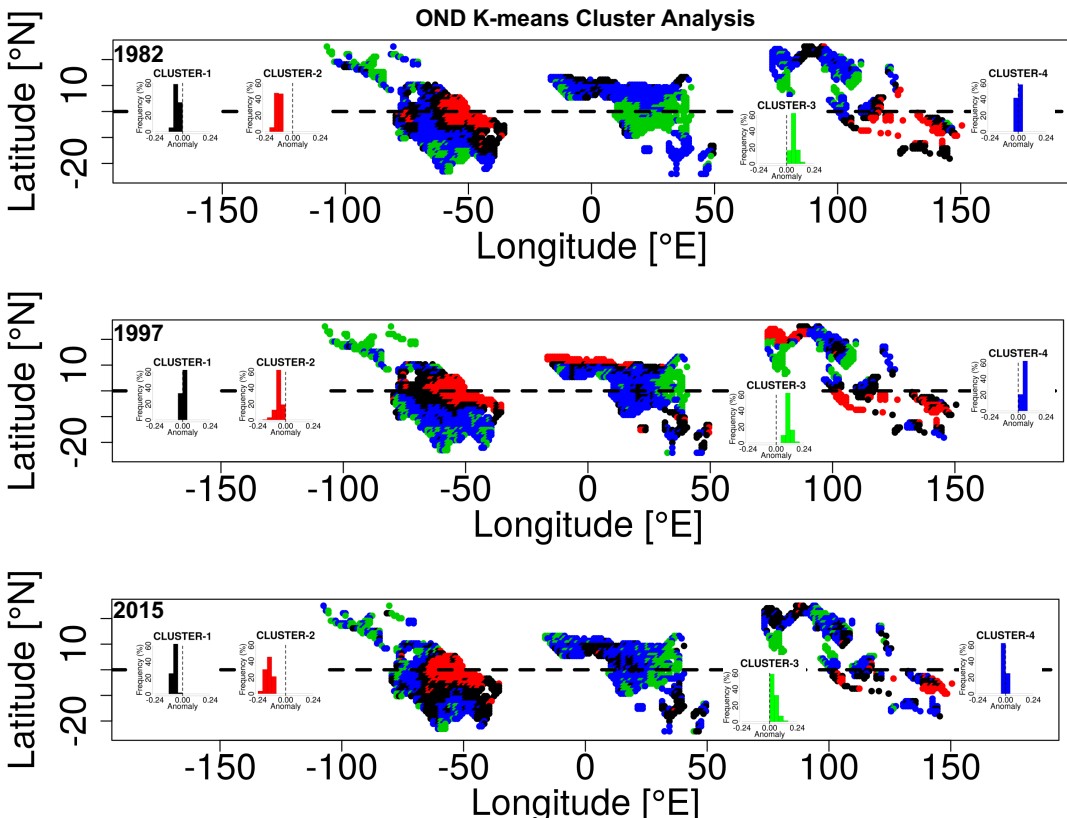

**Figure 6a**: K-means cluster analysis results for October to December (OND) for 1982 (top), 1997 (middle) and 2015 (bottom). Corresponding histograms of soil moisture anomalies for each of the four clusters also shown. Anomalies relative to 1979-2016 period.

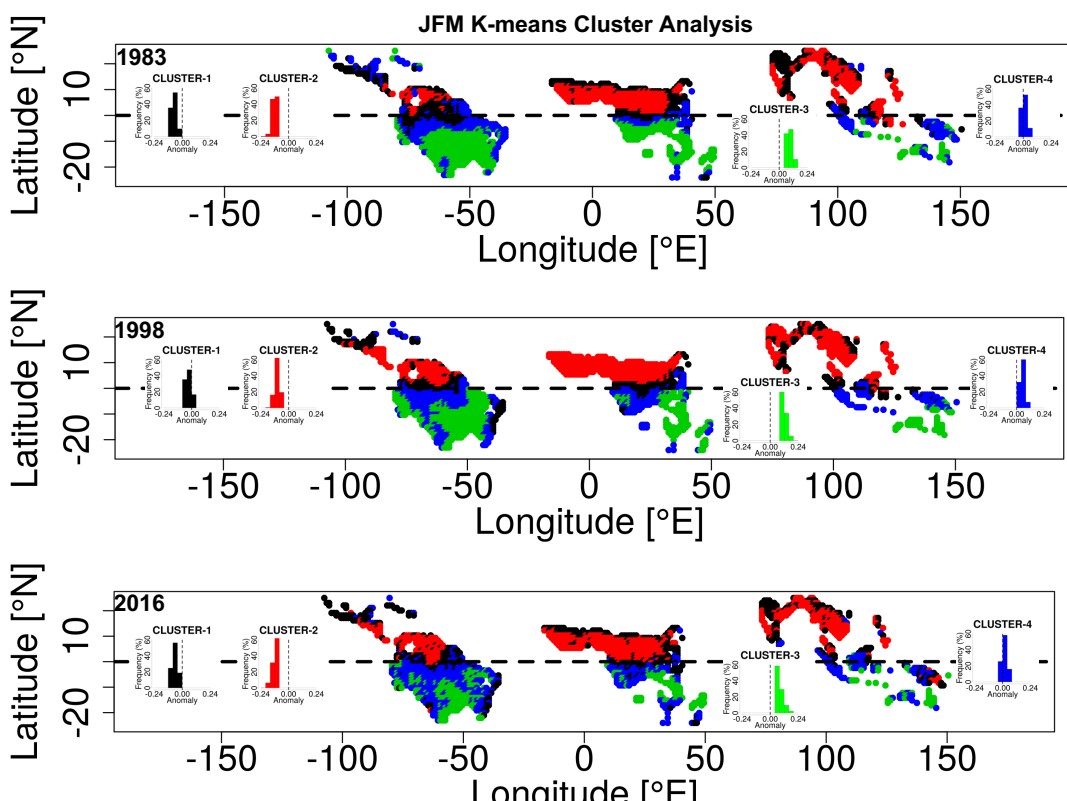

**Figure 6b**: Same as Figure 6a, but for January to March (JFM) in 1983 (top), 1998 (middle) and 2016 (bottom).

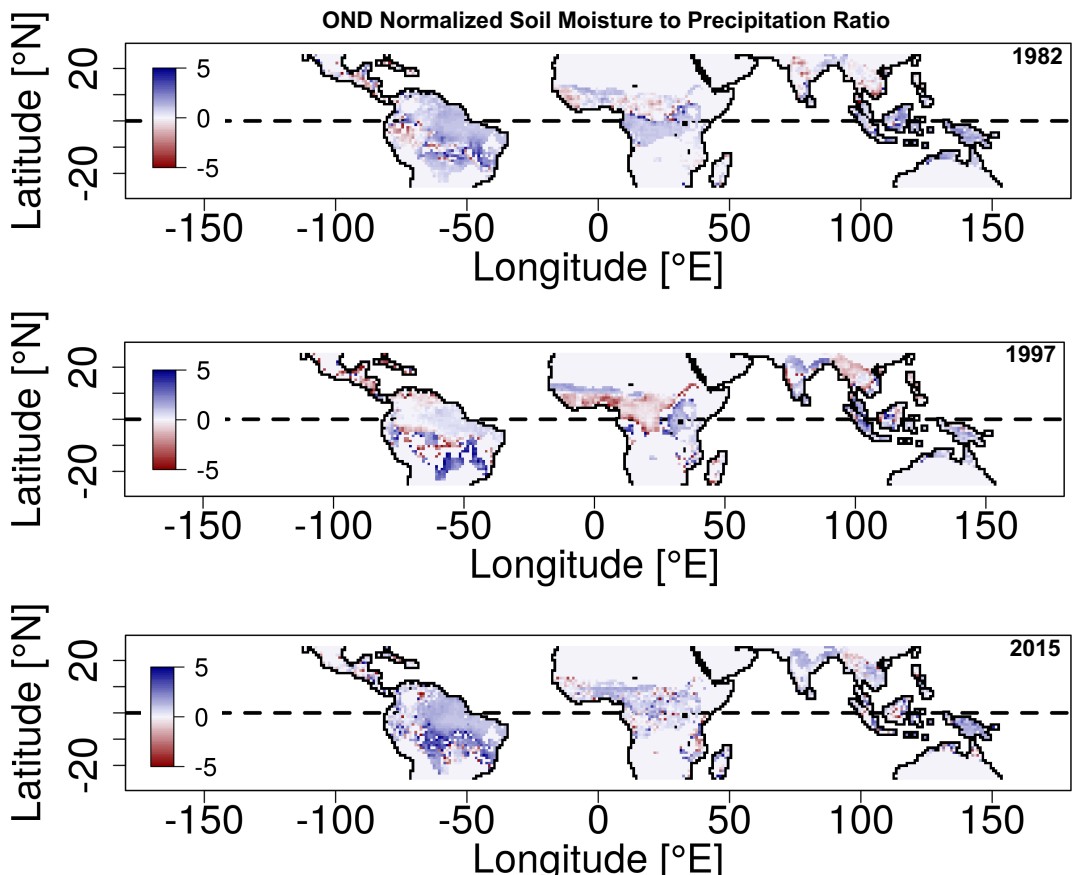

**Figure 7a**: Ratio of GLDAS soil moisture to precipitation change computed using October to December (OND) anomalies during El Niño years 1982-83, 1997-98, and 2015-16 relative to previous years. Anomalies normalized by the mean relative to 1979-2016 period.

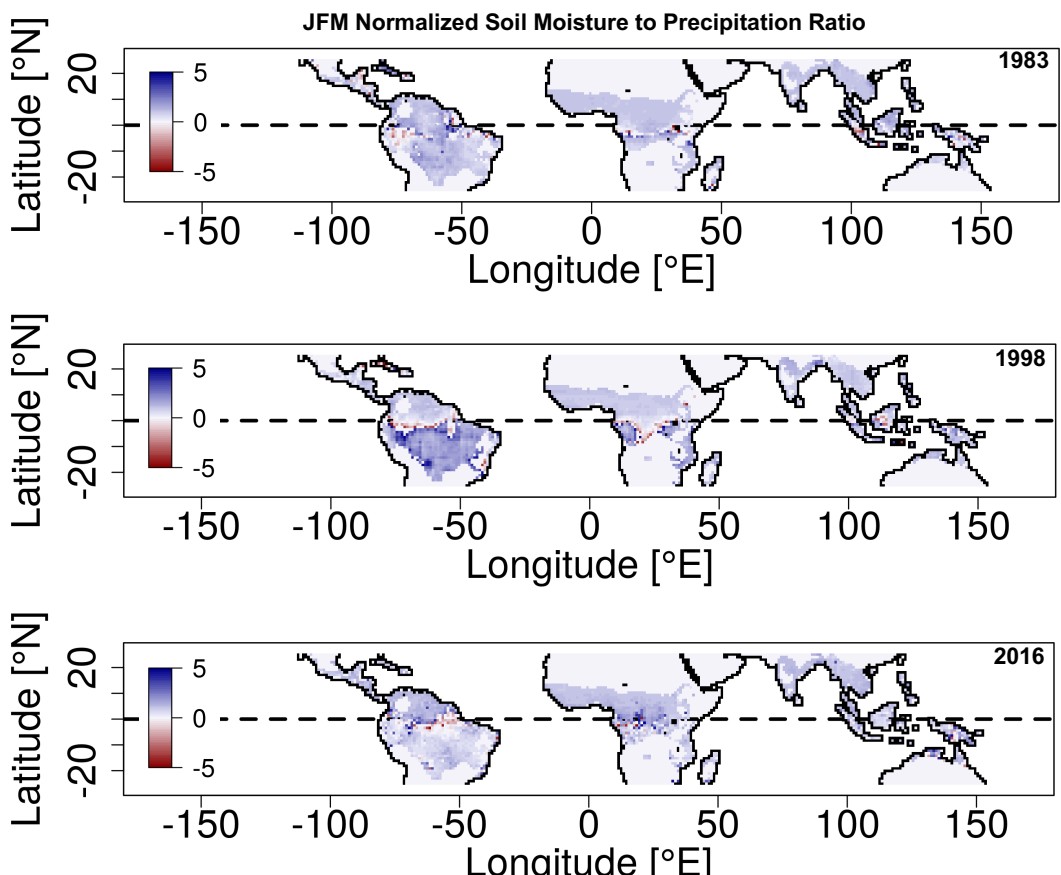

**Figure 7b**: Same as Figure 7a but for January to March in 1983 (top), 1998 (middle) and 2016 (bottom).



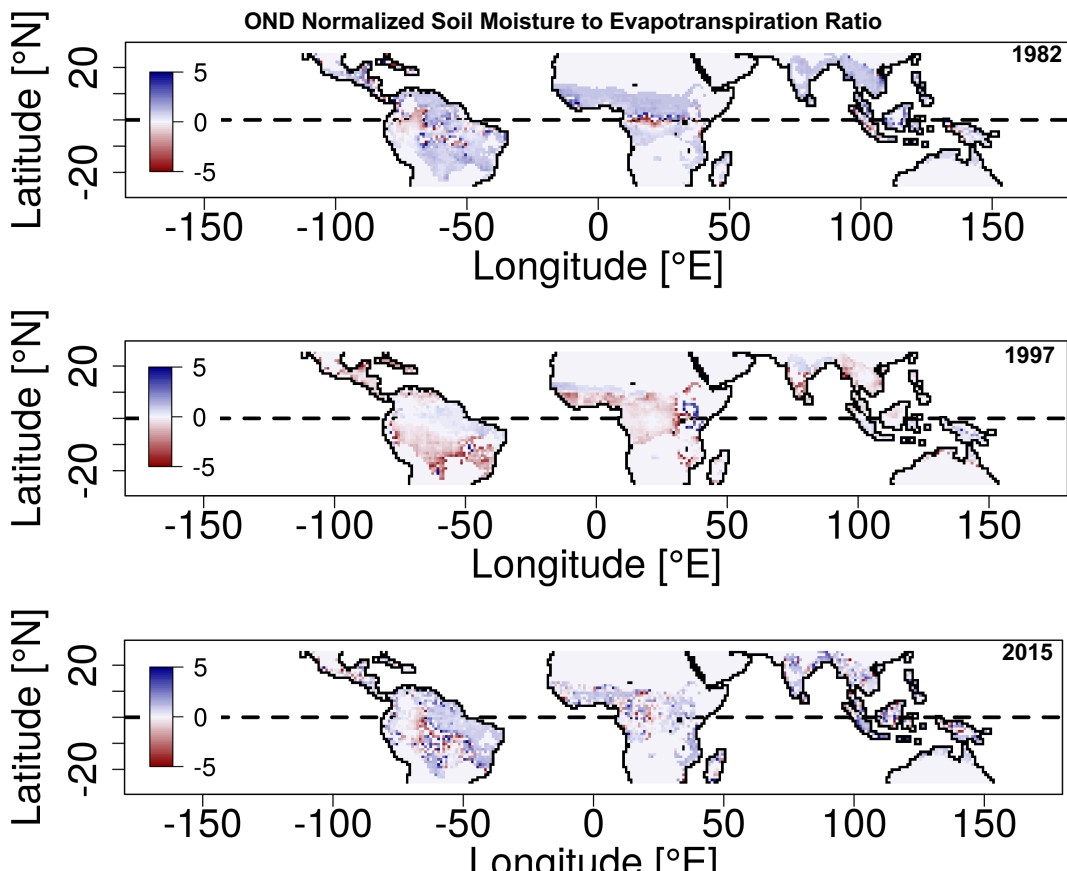

**Figure 8a**: Ratio of GLDAS soil moisture to evapotranspiration change computed using October to December (OND) anomalies during El Niño years 1982-83, 1997-98, and 2015-16 relative to previous years. Anomalies normalized by the mean relative to 1979-2016 period.

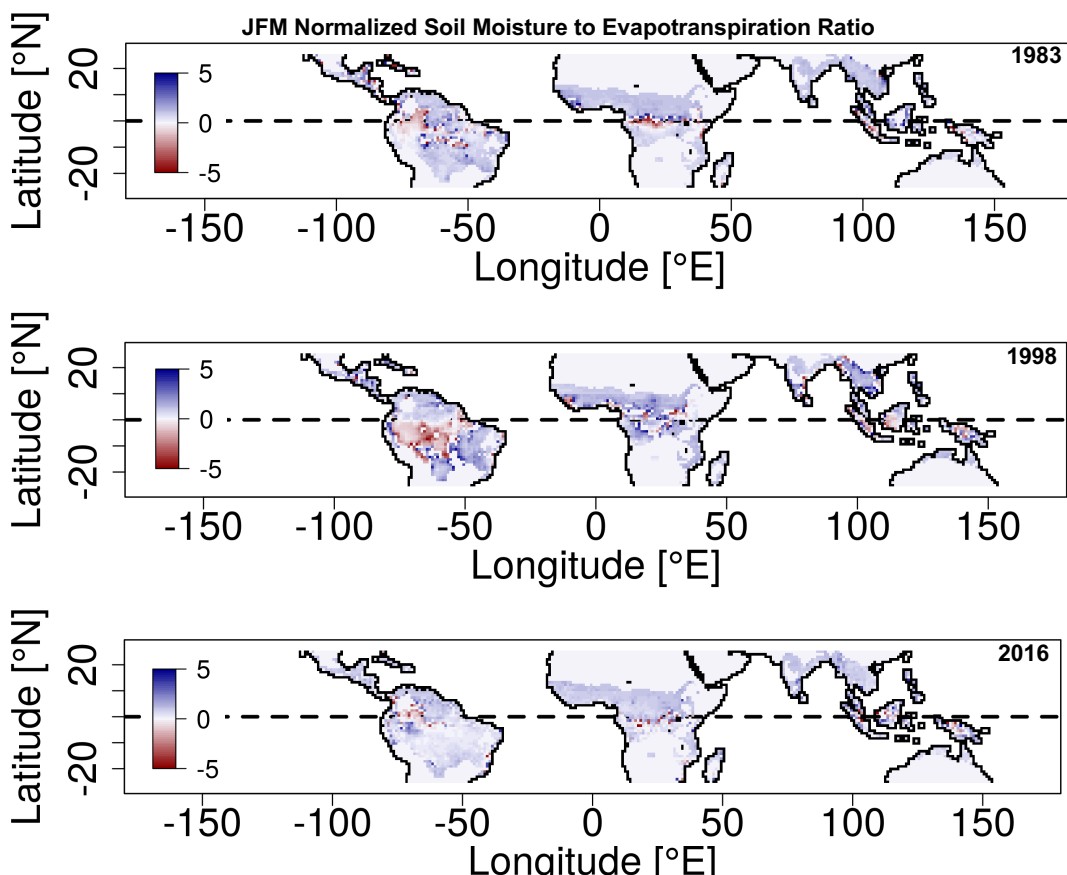

**Figure 8b**: Same as Figure 8a but for January to March in 1983 (top), 1998 (middle) and 2016 (bottom).
Anomalies normalized by the mean relative to 1979-2016 period.





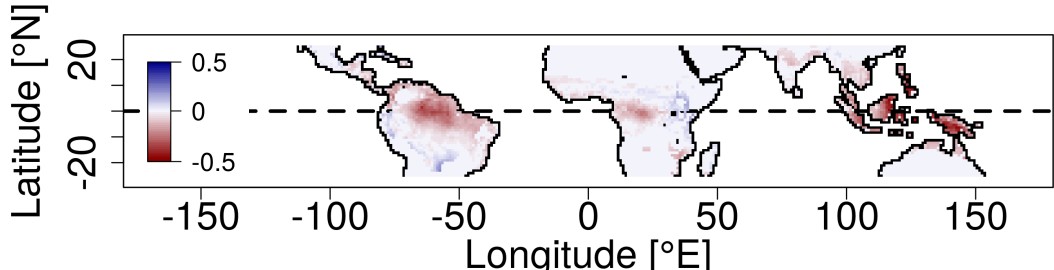

**Figure 9**: Pearson correlation coefficient between GLDAS soil moisture and NINO3.4 index from 1979 to 2016. Colors indicate regions where the mean correlation was negative (red) and positive (blue).



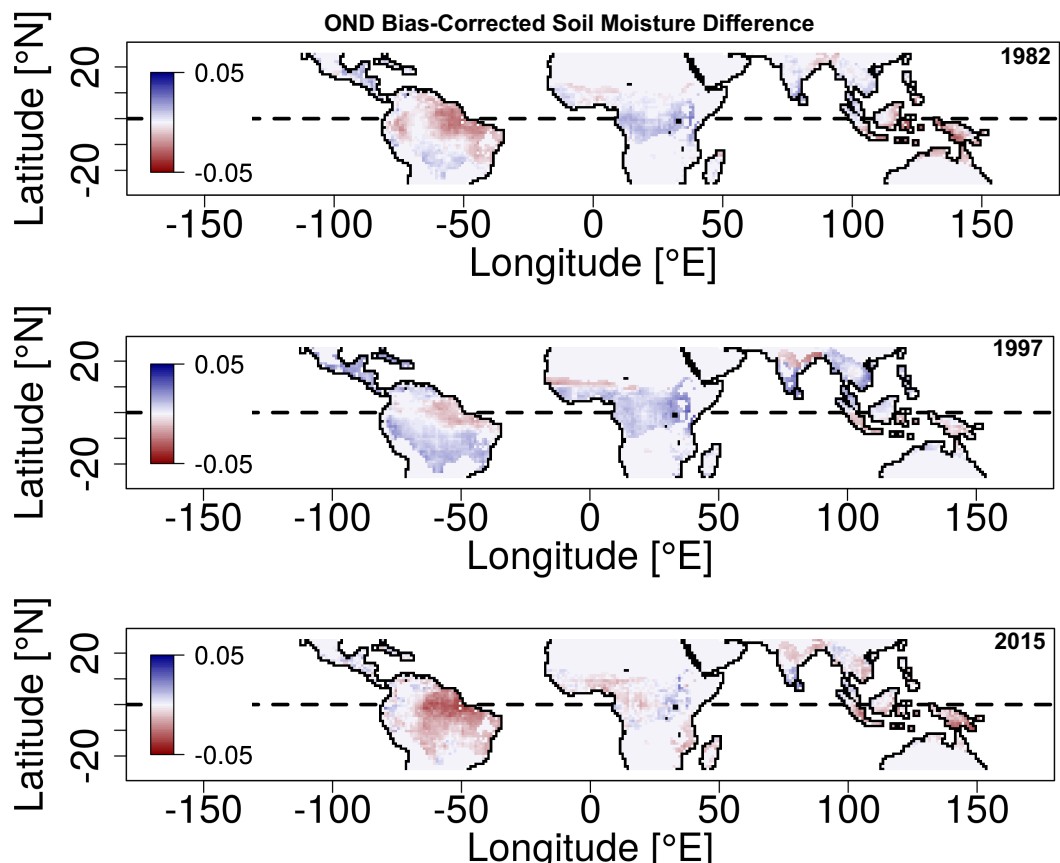

**Figure 10a**: Difference in October to December (OND) change in GLDAS soil moisture anomalies when bias correction is applied relative to no bias correction. Red pixels indicate regions that showed a decrease, while blue regions indicate regions that showed an increase with the addition of the bias correction. Plots shown for the super El Niño years 1982 (top), 1997 (middle), and 2015 (bottom).



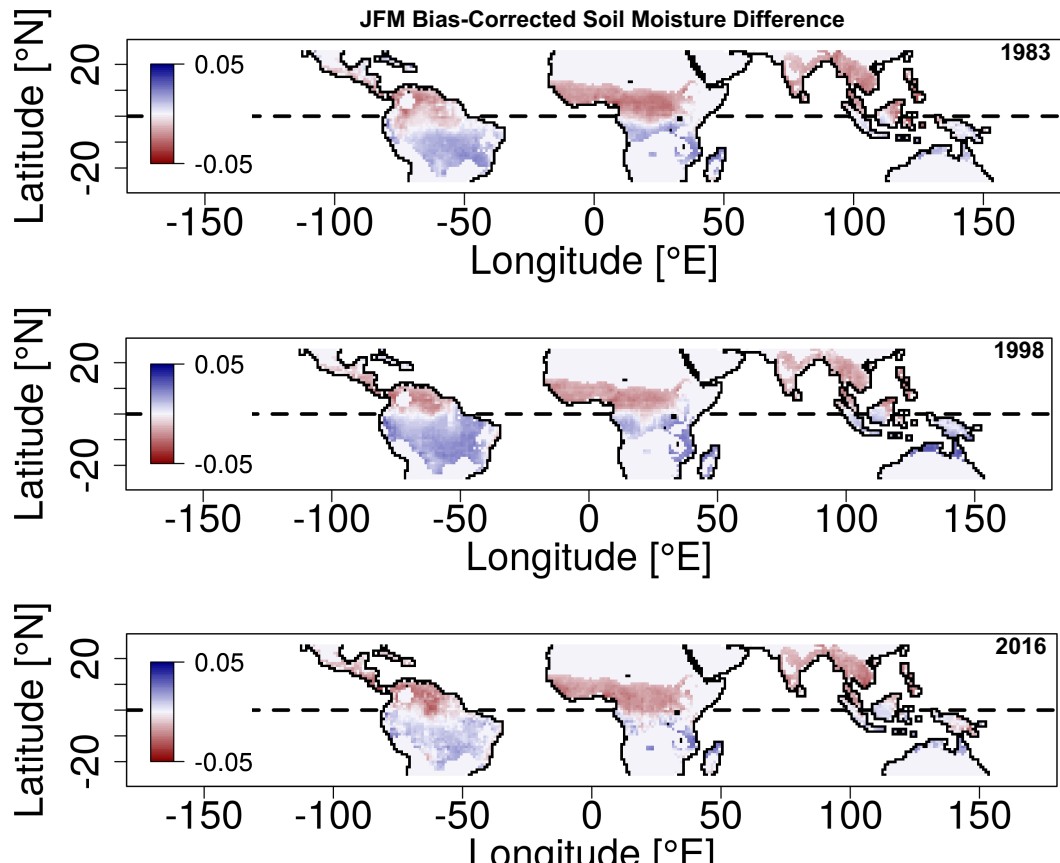

**Figure 10b**: Same as Figure 10a but for January to March (JFM) in 1983 (top), 1998 (middle) and 2016 (bottom).