# Peer review of "The pan-tropical response of soil moisture to El Niño"

_Hydrology and Earth System Sciences, 2019_

## Referee Comment (RC1) · Anonymous Referee #1 · 23 Dec 2019

The paper aims to update the documentation of the response of soil moisture to El Niño in tropical regions. The work is very well presented and has a high potential to be published because of its contribution to improve the predictions of the impacts of extreme El Niño on tropics hydrology. The scientific methods and assumptions are valid. However, I give some general comments for strengthen the results and a second part I will give to the authors some minor comments concerning the manuscript presentation.

General Comments:

The work seems to be a product test-oriented paper, knowing the need to evaluate remote sensing products for large-scale phenomena as El Niño. Even being an inter-

esting and ambitious paper, I recommend to contextualize it into the current problems of the large-scale hydroclimatology of extreme events and how valid is the comparison with in-situ data for explaining the relationship between spatial scales (please reinforce the paragraph on page 10, lines 238-247).

For instance, It is well known that a significant fraction of precipitation above land originates from local evaporation because of soil moisture content. Therefore, which role soil moisture plays on extreme El Niño precipitation apart from oceanic influence? It could be interesting to infer in which regions soil moisture content could be important as sea surface temperature for regional climate with GLDAS product. There is a soil-moisture-rainfall feedback mechanism shown in Figures 6, 7 and 8, please explain and linking them with more detail, taking into account the non stationarity of Extreme El Niño events.

Minor Comments:

-Please add the location of 16 observation sites into a figure (maybe in Figure 9 or another).

In References section:

-Please add in the manuscript or delete the next references: Ashok et al., 2007; Banholzer et al., 2014; Kranowski and Lai,1998; Kug et al., 2009; Larkin and Harrison, 2005; L'Heureux et al., 2015; Takahashi et al., 2011; Wang et al., 2017; Yu and Kim, 2013.

-Please list in alphabetical order: van Schaik et al., 2018.

-Correct the year: Schaefer et al., 2007; Caliński and Harabasz, 1974.

-List in ascending year, references with the same author: Miralles, D.G; Koster, R.D.

-A missing reference: Alsdorf et al., 2007 (see line 486, page 21).

-Correct the last name: Gaur and Mohanty, 2016 (see line 498, page 21); Ratkowksky

and Lance, 1978 (see page 36).

---

## Referee Comment (RC2) · Anonymous Referee #2 · 30 Jan 2020

The manuscript "The pan-tropical response of soil moisture to El Niño" evaluates the quality of the GLDAS soil moisture dataset and uses GLDAS to investigate response of soil moisture to three recent "super El Niño" events. The study is of interest to a broad scientific community, including researchers in hydrology, ENSO dynamics, and land-atmosphere interaction. The paper is well written. But the current manuscript can be improved in both analysis and paper structure. Therefore I recommend major revisions before being considered for publication in HESS. Major suggestions: 1) On paper structure, although the title implies a scientifically oriented study, namely response in tropical soil moisture to El Niño, the paper emphasizes the technical parts, namely evaluation and bias correction of GLDAS, probably too much. I suggest keeping the title and scientific emphasis of the paper, while merging some of the discussion about

data quality and bias correction into the main results/discussion. For example, it might be more convincing if you use the bias-corrected data in Figures 5-9, since you suggest the bias-corrected data is better than the original in Figure 10. You can keep the figures from original data in supplemental and briefly discuss the difference between the original data and bias-corrected data. 2) On analysis, you might consider an alternative or additional way of doing k-means clustering analysis. Currently, the clustering is done for each case separately. While there is advantage of doing this, the disadvantage is that the spatial distribution of clusters you get varies by each El Niño event, making the comparison of clusters between different events (Tables 3-6) a little apple-to-orange. I suggest you to repeat the k-means clustering analysis with all three events together - a multi-dimension k-means clustering analysis. In this way, you should be able to get better summary of the results, for example you can directly tell which regions have the most robust (consistent) response to all three events. Then readers can clearly see each cluster from all three events, their spatial distribution, their response sign and magnitude. I think it worths trying this way at least.

Minor suggestions: 1) You might consider adding the bias-corrected line to Figure 1. 2) Add continental outlines in Figure 6.
* * *

---

## Author Comment (AC1) · 19 Feb 2020

Review #1: Solander et al. The paper aims to update the documentation of the response of soil moisture to El Niño in tropical regions. The work is very well presented and has a high potential to be published because of its contribution to improve the predictions of the impacts of extreme El Niño on tropics hydrology. The scientific methods and assumptions are valid. However, I give some general comments for strengthen the results and a second part I will give to the authors some minor comments concerning the manuscript presentation.

Author Response: We thank the reviewer for the overall positive review for this manuscript. We address each comment below.

[Figure]

General Comments: The work seems to be a product test-oriented paper, knowing the need to evaluate remote sensing products for large-scale phenomena as El Niño. Even being an interesting and ambitious paper, I recommend to contextualize it into the current problems of the large-scale hydroclimatology of extreme events and how valid is the comparison with in-situ data for explaining the relationship between spatial scales (please reinforce the paragraph on page 10, lines 238-247). For instance, It is well known that a significant fraction of precipitation above land originates from local evaporation because of soil moisture content. Therefore, which role soil moisture plays on extreme El Niño precipitation apart from oceanic influence? It could be interesting to infer in which regions soil moisture content could be important as sea surface temperature for regional climate with GLDAS product. There is a soilmoisture-rainfall feedback mechanism shown in Figures 6, 7 and 8, please explain and linking them with more detail, taking into account the non stationarity of Extreme El Niño events.

Author Response: We have added more discussion to potential soil moisture rainfall feedbacks relevant to what is shown in the revision of the manuscript (see below).

The spatial patterns exhibited in Figures 7 and 8 highlight some important soil moisture feedbacks during El Niño that may be related to seasonal changes in precipitation recycling, which is known to be a particularly important process for moisture generation over the Amazon (Eltahir and Bras, 1996). For example, there was a large region over the southern Amazon where precipitation and evapotranspiration were inversely related to soil moisture during OND and the location of this disagreement generally shifted further north towards the equator during JFM. Likewise, over Africa, there was a large region where precipitation and evapotranspiration were inversely related to soil moisture centered north of the equator during OND, but the location of this disagreement shifted south of the equator during JFM. Negative feedbacks among these variables occur either where soils are close to saturation and additional soil moisture is more likely to result in runoff than increases in evapotranspiration and precipitation, or where soils are so dry that additional moisture is less likely to cause a corresponding

increase in evapotranspiration or precipitation due to soil moisture suctioning (Seneviratne et al., 2010; Yang et al., 2018). It is more likely that the latter process is occurring over the Amazon while the former is occurring over equatorial Africa given the seasonal occurrence of dry and wet soil moisture conditions shown over these regions in Figure 5. Moreover, strong El Niños are frequently associated with a negative phase of the Atlantic dipole that displaces the Inter Tropical Convergence Zone northward, which favors drier conditions over the Amazon and wetter conditions over sub-Saharan Africa (Hastenrath and Heller, 1977). The displacement of the ITCZ and Pacific warming in Peru also weakens trade winds over the Amazon, which serves to limit moisture transport from the Atlantic towards the Amazon further drying out this region (Satyamurty et al., 2013). The end result of these changes are negative ratios shown in Figures 7 and 8 potentially highlighting weaker precipitation recycling that shifts north from OND to JFM over the Amazon, but south from OND to JFM over equatorial Africa. When precipitation recycling weakens, a greater proportion of atmospheric moisture over these regions will be derived from further away over the ocean rather than locally over land.

References: Eltahir, E.A.B., and Bras, R. L., 1994, Precipitation recycling in the Amazon basin, Quart. J. Roy. Meteor. Soc., 120, 861-880.

Hastenrath, S., and Heller, L., 1977, Dynamics of climatic hazards in northeast Brazil, Quart. J. R. Met. Soc., 103, 77-92.

Satyamurty, P., da Costa, C. P. W., and Manzi, A. O., 2013, Theor. Appl. Clim., Moisture source for the Amazon Basin: a study of contrasting years, 111, 195-209.

Seneviratne, S.I., and Coauthors, 2010, Investigating soil moisture-climate interactions in a changing climate: A review, Earth-Sci. Rev., 99, 125-161.

Yang, L., Sun, G. Zhi, L., and Zhao, J., 2018, Negative soil moisture-precipitation feedback in dry and wet regions, Sci. Rep., 8, 1-9.

Minor Comments: -Please add the location of 16 observation sites into a figure (maybe

in Figure 9 or another).

Author Response: We have added the location of the 16 observation sites into Figure 5a (see Figure 1 attached), which is the first figure that shows the global tropics map after we discuss integration of the in-situ data.

In References section: -Please add in the manuscript or delete the next references: Ashok et al., 2007; Banholzer et al., 2014; Kranowski and Lai,1998; Kug et al., 2009; Larkin and Harrison, 2005; L'Heureux et al., 2015; Takahashi et al., 2011; Wang et al., 2017; Yu and Kim, 2013.

Author Response: These references were relevant to an earlier draft of the manuscript that has since been removed. We have removed these references from the manuscript.

-Please list in alphabetical order: van Schaik et al., 2018.

Author Response: We have moved this reference to correct alphabetical order.

-Correct the year: Schaefer et al., 2007; Cali ′ nski and Harabasz, 1974.

Author Response: We have corrected the years for these references in the reference list.

-List in ascending year, references with the same author: Miralles, D.G; Koster, R.D.

Author Response: We have reordered these same-author references to ascending year.

-A missing reference: Alsdorf et al., 2007 (see line 486, page 21).

Author Response: We have added this reference to the reference list.

-Correct the last name: Gaur and Mohanty, 2016 (see line 498, page 21); Ratkowksky and Lance, 1978 (see page 36).

Author Response: We have corrected the last names of these citations.

[Figure]

**OND Soil Moisture Anomalies**

[Figure]

**Figure 1**: October to December (OND) change in bias-corrected GLDAS soil moisture anomalies during the super El Niño years 1982 (top), 1997 (middle), and 2015 (bottom) relative to the previous years. Anomalies relative to 1979-2016 period. Green circles represent 16 in-situ data sample locations.

**Fig. 1.**

---

## Author Comment (AC2) · 19 Feb 2020

Review #2: The manuscript "The pan-tropical response of soil moisture to El Niño" evaluates the quality of the GLDAS soil moisture dataset and uses GLDAS to investigate response of soil moisture to three recent "super El Niño" events. The study is of interest to a broad scientific community, including researchers in hydrology, ENSO dynamics, and land-atmosphere interaction. The paper is well written. But the current manuscript can be improved in both analysis and paper structure. Therefore I recommend major revisions before being considered for publication in HESS.

Author Response: We thank the reviewer for the overall positive review. We agree that the analysis and paper structure can be improved. We address each comment below.

[Figure]

Major suggestions: 1) On paper structure, although the title implies a scientifically oriented study, namely response in tropical soil moisture to El Niño, the paper emphasizes the technical parts, namely evaluation and bias correction of GLDAS, probably too much. I suggest keeping the title and scientific emphasis of the paper, while merging some of the discussion about data quality and bias correction into the main results/discussion. For example, it might be more convincing if you use the bias-corrected data in Figures 5-9, since you suggest the bias-corrected data is better than the original in Figure 10. You can keep the figures from original data in supplemental and briefly discuss the difference between the original data and bias-corrected data.

Author Response: We have re-done Figures 5-9 using bias-corrected soil moisture estimates (see attached Figures 1-9) and correspondingly removed the old Figure 10 from the manuscript. We have moved the old Figures 5-9 generated from the non-biased GLDAS data to the Supplemental. Also, we have moved some of the discussion about data quality and bias correction to the main results (e.g. Lines 452-455 & 472-480).

2) On analysis, you might consider an alternative or additional way of doing k-means clustering analysis. Currently, the clustering is done for each case separately. While there is advantage of doing this, the disadvantage is that the spatial distribution of clusters you get varies by each El Niño event, making the comparison of clusters between different events (Tables 3-6) a little apple-to-orange. I suggest you to repeat the k-means clustering analysis with all three events together - a multi-dimension k-means clustering analysis. In this way, you should be able to get better summary of the results, for example you can directly tell which regions have the most robust (consistent) response to all three events. Then readers can clearly see each cluster from all three events, their spatial distribution, their response sign and magnitude. I think it worths trying this way at least.

Author Response: We agree and have conducted a multi-dimensional analysis in Figure 6 by showing the overlap of existing clusters of the three major El Niño events for

each season. This was achieved by determining the locations where there was a match in the cluster group over the three major El Niño events (see attached Figures 3-4) and then calculating additional statistics within Tables 3-6 (see attached Figure 10).

Minor suggestions: 1) You might consider adding the bias-corrected line to Figure 1.

Author Response: We have replaced the old GLDAS data with the bias-corrected line in Figure 1 (see attached Figure 11).

2) Add continental outlines in Figure 6.

Author Response: We have added the continental outlines in Figure 6 (see attached Figures 3-4).

———————————————

[Figure]

**OND Soil Moisture Anomalies**

[revised manuscript text omitted]